# Evanescent waves modulate energy efficiency of photocatalysis within TiO$_2$ coated optical fibers illuminated using LEDs

Yinghao Song [1,4], Li Ling [1,4,5✉], Paul Westerhoff [2] & Chii Shang [1,3,5✉]

Coupling photocatalyst-coated optical fibers (P-OFs) with LEDs shows potential in environmental applications. Here we report a strategy to maximize P-OF light usage and quantify interactions between two forms of light energy (refracted light and evanescent waves) and surface-coated photocatalysts. Different TiO$_2$-coated quartz optical fibers (TiO$_2$-QOFs) are synthesized and characterized. An energy balance model is then developed by correlating different nano-size TiO$_2$ coating structures with light propagation modes in TiO$_2$-QOFs. By reducing TiO$_2$ patchiness on optical fibers to 0.034 cm$^2$/cm$^2$ and increasing the average interspace distance between fiber surfaces and TiO$_2$ coating layers to 114.3 nm, refraction is largely reduced when light is launched into TiO$_2$-QOFs, and 91% of light propagated on the fiber surface is evanescent waves. 24% of the generated evanescent waves are not absorbed by nano-TiO$_2$ and returned to optical fibers, thus increasing the quantum yield during degradation of a refractory pollutant (carbamazepine) in water by 32%. Our model also predicts that extending the TiO$_2$-QOF length could fully use the returned light to double the carbamazepine degradation and quantum yield. Therefore, maximizing evanescent waves to activate photocatalysts by controlling photocatalyst coating structures emerges as an effective strategy to improve light usage in photocatalysis.

[1] Department of Civil and Environmental Engineering, The Hong Kong University of Science and Technology, Hong Kong, China. [2] School of Sustainable Engineering and the Built Environment and Nanosystems Engineering Research Center for Nanotechnology-Enabled Water Treatment (NEWT), Arizona State University, Tempe, AZ, USA. [3] Hong Kong Branch of Chinese National Engineering Research Center for Control & Treatment of Heavy Metal Pollution, The Hong Kong University of Science and Technology, Hong Kong, China. [4] These authors contributed equally: Yinghao Song, Li Ling. [5] These authors jointly supervised this work: Li Ling, Chii Shang. ✉email: celingli@ust.hk; cechii@ust.hk

Heterogeneous photocatalysis has shown tremendous potential for solving environmental problems and reducing the energy crisis in the past few decades. Its applications include pollutant degradation, bacteria, and virus inactivation, water splitting for energy production, organic transformations, and more[1–6]. Upon absorption of light, photocatalysts generate hole-electron ($h^+$-$e^-$) pairs that subsequently undergo redox reactions with molecules adsorbed on photocatalyst surfaces[7–9]. However, current reactor configurations compromise the viability of photocatalytic processes because both slurry and fixed-bed reactors scatter and occlude light, consequently reducing the overall system energy efficiency[10–13].

Launching light from energy-efficient and low-cost LEDs into optical fibers that are coated with photocatalysts is a new photocatalytic reactor design[14–18]. Such design shows dramatic improvements in quantum yields during pollutant degradation (i.e., moles of pollutants degraded per mole of photons absorbed by photocatalysts). As light propagates along photocatalyst-coated optical fibers (P-OFs), a fraction of the light is refracted from the lower refractive index optical fibers into the higher refractive index surface-coated photocatalyst layers (Supplementary Note 1). This produces reactive oxygen species (ROS), which degrade pollutants at the interface between photocatalysts and water containing the pollutants[17,19,20]. This configuration minimizes light loss due to scattering and occlusion and nearly doubles quantum yields of degradation of three pollutants when compared against slurry reactor systems at equivalent photocatalyst masses[17,20,21]. However, due to thick and dense photocatalyst layers dip-coated on optical fibers using highly concentrated suspension of photocatalysts in previous reports, most light delivered to the P-OFs refracts out of the fiber near the beginning sections of the coating[19,20,22,23]. As such, the majority of the refracted light escaped from the P-OFs without activating photocatalysts and lost its energy in water. Maximizing refracted light absorption by coating more photocatalysts improves light utilization, but thicker and more dense photocatalyst coating layers limited mass transfer of pollutants from water to ROS-producing sites[24]. The resulting quantum yields were thus not enhanced[20]. To realize the potential of reactor designs that can

efficiently leverage LED technological advances, P-OFs achieving higher light utilization without compromising photocatalytic performance need to be developed.

To date, P-OF research has focused on managing refracted light. However, light has two energy forms when propagating along P-OFs: refracted light and evanescent waves. We hypothesized that maximizing evanescent wave energy, which is generated during total internal reflection (TIR) (details in Supplementary Note 2) is a better strategy to activate surface-coated photocatalysts than managing refracted light energy. Unlike refracted light, which propagates away from optical fibers and loses its energy in water, evanescent waves propagate on optical fiber surfaces. Evanescent wave energy that does not react with the photocatalyst returns and continues to propagate along the optical fiber where it can subsequently react with photocatalysts located further along the axial path of the optical fiber[25,26]. Thus, light reacts more efficiently with surface-coated photocatalysts through evanescent waves than refracted light.

In this work, to test our hypothesis, we evaluate the ability of TiO$_2$-coated quartz optical fibers (TiO$_2$-QOFs), which have novel tunable surface "patches" of TiO$_2$ layers, to promote evanescent wave generation to activate TiO$_2$ (Fig. 1a) and degrade a refractory pollutant (carbamazepine) by photocatalytic generated hydroxyl radicals (HO•) in bulk solution and pores in TiO$_2$ coating layers (Fig. 1b). Compared with 6.5-cm TiO$_2$-QOFs coated with thick and dense TiO$_2$ layers, the 6.5-cm TiO$_2$-QOFs coated with lower TiO$_2$ patchiness increase quantum yield by 32% without compromising carbamazepine degradation efficiency. The TiO$_2$-QOFs coated with low TiO$_2$ patchiness are fabricated by dip-coating 4.8 μg/cm$^2$ nano-TiO$_2$ (P25, anatase/rutile 85/15, particle size 21 nm), which is selected for its wide use[7], on the surfaces of quartz optical fibers, which are selected for their high UV transmission[27]. This coating strategy leaves only 3.4% of the optical fiber surface in direct contact with TiO$_2$ and creates 114.3 nm, on-average, interspace distance between the fiber surfaces and the coated TiO$_2$ layers. These coating structures are confirmed by scanning electron microscopy (SEM) and transmission electron microscopy (TEM). Based on the characterized TiO$_2$ coating structures, an energy balance model is

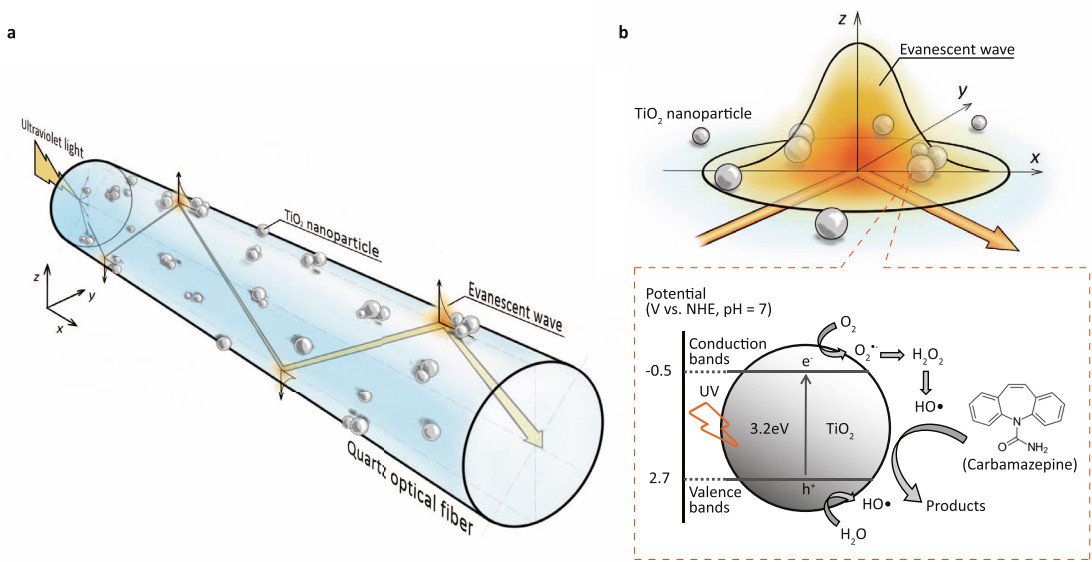

**Fig. 1 Schematics of the TiO$_2$-QOFs with tunable surface "patches" of TiO$_2$ layers. a** Light propagates along the TiO$_2$-QOFs and generates evanescent waves when light strikes the optical fiber surfaces without TiO$_2$ nanoparticles; and **b** TiO$_2$ nanoparticle activation by the generated evanescent waves, and the subsequent production of hydroxyl radicals (HO•) to degrade carbamazepine.

**Table 1 Dip-coating conditions and TiO$_2$ coating layer parameters in the three TiO$_2$-QOFs.**

| ID | Dip-coating conc. (mg/L) | Dipping duration (min) | Coating/drying cycles / | Area-specific TiO$_2$ coating density (µg/cm$^2$) | TiO$_2$ coating porosity / |
|---|---|---|---|---|---|
| TiO$_2$-QOF-High | 10,000 | 0.5 | 5 | 81.8 ± 5.1 | 62.2% |
| TiO$_2$-QOF-Med | 10,000 | 0.5 | 1 | 16.9 ± 2.5 | 71.0% |
| TiO$_2$-QOF-Low | 40 | 60 | 1 | 4.8 ± 0.6 | 89.6% |

developed to describe the light propagation in the TiO$_2$-QOFs as functions of the TiO$_2$ coating layer structure parameters. From the experiments and model, we find that 91% of the radiant energy delivered to the low patchiness TiO$_2$-QOFs propagates on the fiber surfaces as evanescent waves, and the behavior of evanescent waves interacting with the TiO$_2$ coating results in a saving of 23% for the radiant energy delivered to the TiO$_2$-QOFs. Extending the low patchiness TiO$_2$-QOFs from 6.5 to 26 cm is even more beneficial, achieving about 2× carbamazepine degradation and also doubling quantum yields. In addition, the low patchiness coating on the 26 cm fiber uses 77% fewer photocatalysts than the TiO$_2$-QOFs with thick and dense TiO$_2$ coating layers. Thus, modulating TiO$_2$ patchiness on the fiber surface emerges as a tunable parameter to optimize evanescent wave energy rather than refracted light energy to improve both quantum yield and pollutant degradation.

## Results and discussion

**Characterization and analysis of TiO$_2$ coating layers.** We fabricated three TiO$_2$-QOFs (photos are shown in Supplementary Fig. 3) by dipping quartz optical fibers into P25 suspensions under different conditions[19,20] (Table 1). They were labeled as TiO$_2$-QOF-High, TiO$_2$-QOF-Med, and TiO$_2$-QOF-Low, which had 81.8, 16.9, and 4.8 µg TiO$_2$ per cm$^2$ of optical fiber, respectively. Electron microscopy revealed the exterior surface and cross-sectional morphologies of the uncoated optical fiber and the three TiO$_2$-QOFs (Fig. 2a–h). As shown in the SEM images (Fig. 2a–d), the exterior surface of the uncoated optical fiber has no TiO$_2$ attached (Fig. 2a), while the TiO$_2$-QOFs had porous TiO$_2$ coating layers. TiO$_2$-QOF-High had the densest TiO$_2$ coating layer (porosity of 62% (Fig. 2b)). TiO$_2$-QOF-Med and TiO$_2$-QOF-Low were more porous (71% and 90%, respectively) (Fig. 2c and d). The porous structures of TiO$_2$ coating layers were also confirmed by atomic force microscopy (Supplementary Fig. 4). The cross-sections of the uncoated optical fiber and the three TiO$_2$-QOFs were prepared by cutting them with a focused ion beam system, and their images (Fig. 2e–h) were obtained by TEM. As shown in Fig. 2e, there was no TiO$_2$ attached to the uncoated optical fiber. On TiO$_2$-QOF-High (Fig. 2f), TiO$_2$ nanoparticles formed a porous TiO$_2$ coating layer on the fiber surface. Around 56% of the optical fiber surface had direct contact with TiO$_2$ nanoparticles, and interspaces were created between the optical fiber surface and the TiO$_2$ coating layers. We define "patchiness" as the ratio (cm$^2$/cm$^2$) of the optical fiber surface area with direct TiO$_2$ contact to the total optical fiber surface area. The calculated patchiness of TiO$_2$-QOF-High, TiO$_2$-QOF-Med, and TiO$_2$-QOF-Low were 56%, 25%, and 8%, respectively (Fig. 2f–h). The decreasing patchiness from TiO$_2$-QOF-High to TiO$_2$-QOF-Low was also confirmed by a 3D optical profiler (Supplementary Fig. 6).

Microscopy also allowed us to conceptualize the porous TiO$_2$ coating layers on surfaces of optical fibers as shown in Fig. 2i. The interface is either (i) between the quartz fiber surface and the directly-contact TiO$_2$ (termed quartz/TiO$_2$ interface) or (ii) between the quartz fiber surface and water diffused from the bulk solution in clusters of TiO$_2$ nanoparticles located adjacent to the quartz surface and within the porous coating (termed quartz/water/TiO$_2$ interface). Based on the two interfaces, three modes of light propagation are proposed. In Mode 1, as light approaches the quartz/TiO$_2$ interface, a large portion of radiant energy refracts into TiO$_2$ and a small portion reflects back to quartz due to the higher refractive index of TiO$_2$ than quartz. The amount of radiant energy transmitted into TiO$_2$ and reflected to quartz is determined by Fresnel equation[26,28] and is a function of the incident angle ($\theta$) of light propagating through the fiber and interacting at the fiber surface, which is the angle between incident light and normal (Fig. 2i). In Mode 2, when light approaches the quartz/water/TiO$_2$ interfaces ($\theta <$ the critical angle of TIR ($\theta_c$)), radiant energy is also largely refracted out, and a small portion reflects. In Mode 3, when light approaches the quartz/water/TiO$_2$ interfaces ($\theta > \theta_c$), TIR occurs and evanescent waves are generated. Evanescent waves propagate on the surface of optical fibers with a portion being absorbed by TiO$_2$. The energy that was not absorbed returns to the optical fibers, making Mode 3 utilize light more efficiently than Modes 1 and 2. To promote Mode 3 of light propagation and generate more evanescent waves, it is desirable to have low TiO$_2$ patchiness on the optical fibers. As shown in Fig. 2e–h, TiO$_2$-QOF-Low met this requirement.

**Radiant energy dissipation in TiO$_2$-QOFs.** To verify whether evanescent waves existed in our fabricated TiO$_2$-QOFs when launching light to the fiber core from an LED, we measured the radiant energy dissipated in TiO$_2$-QOFs ($E_{dis}$) at different area-specific TiO$_2$ coating densities (µg/cm$^2$) (Fig. 3a). $E_{dis}$ is defined as the radiant energy delivered into but not transmitted through the TiO$_2$-QOFs (see "Methods" for details), when TiO$_2$-QOFs are exposed to either of two different media (air or water). TiO$_2$-QOF-High at 81.8 µg/cm$^2$ TiO$_2$ showed no statistical difference ($p > 0.05$) in $E_{dis}$ between in air and in water. TiO$_2$-QOF-Med at 16.9 µg/cm$^2$ TiO$_2$ and TiO$_2$-QOF-Low at 4.8 µg/cm$^2$ TiO$_2$ showed a higher $E_{dis}$ in water than that in air ($p < 0.05$), and the differences were 16% and 98%, respectively.

The higher $E_{dis}$ in water was due to the involvement of evanescent waves in TiO$_2$-QOFs. Based on the three modes of light propagation illustrated in Fig. 2i, we surmised that $E_{dis}$ was caused by the two forms of energy (evanescent waves and refracted light). As the penetration depth of evanescent waves ($\Lambda$) is higher when going from quartz to water than from quartz to air, the radiant energy of evanescent waves dissipated in TiO$_2$-QOFs ($E_{E,dis}$) varies in either of the two media as shown in Eqs. (1) and (2)[29–31].

$$\Lambda = \frac{\lambda}{4\pi}\left(n_q^2 \sin^2\theta - n_e^2\right)^{-1/2} \tag{1}$$

$$E_{E,dis} = E_i e^{-\frac{z_n}{\Lambda}} \tag{2}$$

where $\lambda$ is the wavelength of light, $n_q$ and $n_e$ are the refractive index of quartz and external medium, respectively, $E_i$ is the radiant energy of incident light, and $z_n$ is the normalized distance between the fiber surface and TiO$_2$ nanoparticles at a TIR spot, which is equal to the average distance from the center of an

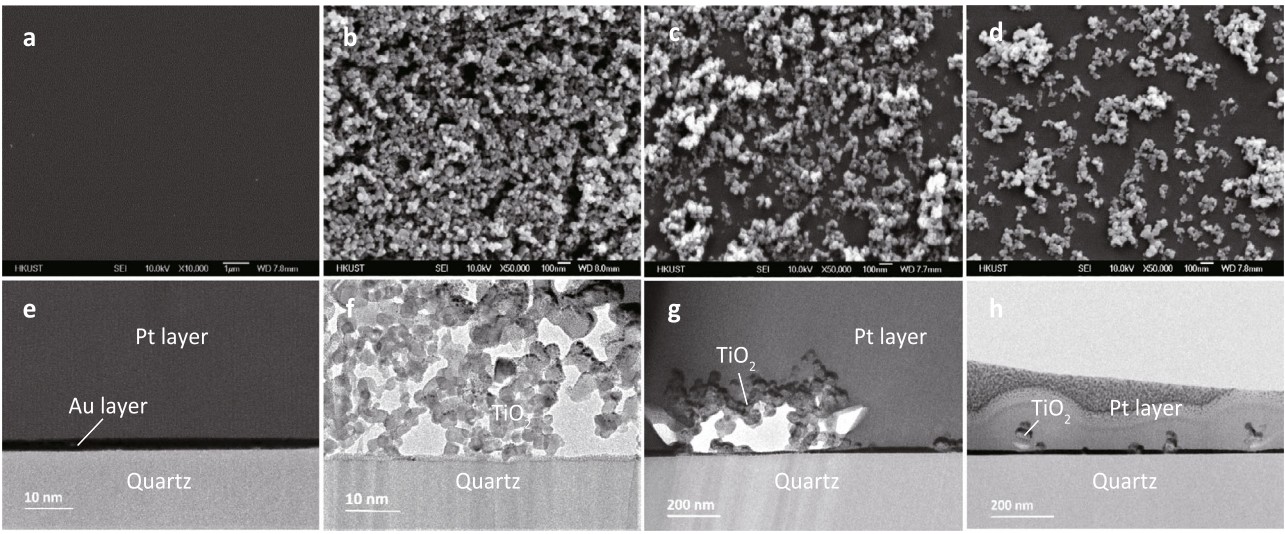

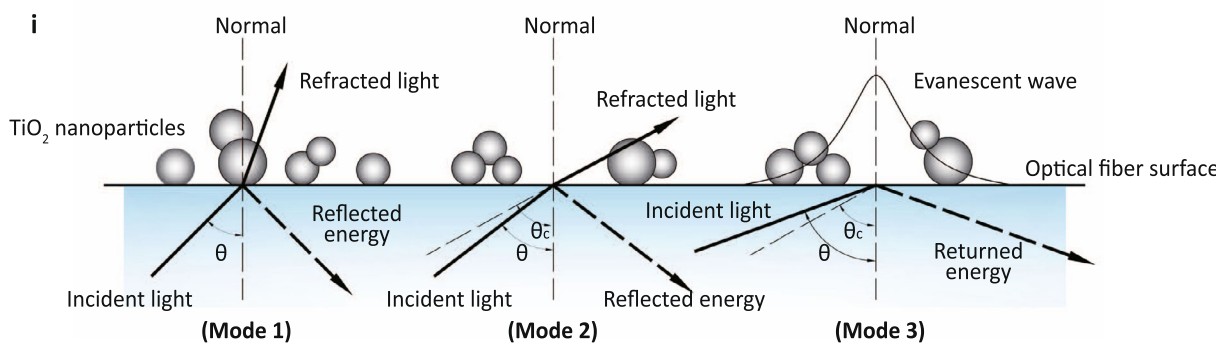

**Fig. 2 Characterization and analysis of the uncoated quartz optical fibers and three TiO₂-QOFs. a–d** Scanning electron microscope (SEM) surface images of the uncoated optical fiber, TiO₂-QOF-High, TiO₂-QOF-Med, and TiO₂-QOF-Low, respectively; **e–h** transmission electron microscopy (TEM) images of cross-sectional morphologies of the uncoated optical fiber, TiO₂-QOF-High, TiO₂-QOF-Med, and TiO₂-QOF-Low, respectively, in which an Au layer was deposited to capture images during the focus ion beam operation and a Pt layer was deposited to protect the TiO₂ coating layers from damaging by focus ion beams; and **i** conceptualized images of TiO₂ coating layers on the quartz optical fiber surfaces and proposed three light propagation modes at the interface.

evanescent field to the closest surrounding TiO₂ nanoparticles within the evanescent field ($\delta_i$), as shown in Fig. 3b[32,33]. In our study, the LED lamp source was fixed at one end of a quartz optical fiber, so $\theta$ remained the same and was always between $0.376\pi$ and $0.495\pi$ (calculation shown in Supplementary Note 5A). $E_i$ and $\lambda$ are inherent properties of the LED, $n_q$ is an inherent property of quartz optical fibers, and $z_n$ is a structural property of the TiO₂ coating layer. As such, these values did not change during the experiments. The $n_e$ of water is higher than the $n_e$ of air (1.33 versus 1.00), so evanescent waves penetrate further from quartz to water than to air suggested by Eq. (1). Therefore, $E_{E,dis}$ in water is higher than $E_{E,dis}$ in air as given by Eq. (2).

Unlike in evanescence wave energy, the radiant energy of refracted light dissipated in TiO₂-QOFs ($E_{R,dis}$) is constant in the two media. At the quartz/air or quartz/water interface, $\theta$ was always greater than $\theta_c$ of TIR, which suggests no mode 2 of light propagation and no radiant energy refracted from quartz to air or from quartz to water. $E_{R,dis}$ only occurred on the quartz/TiO₂ interface, calculated using Fresnel equation shown in Eq. (3):

$$E_{R,dis} = E_i \left\{ 1 - \frac{1}{2} \left\{ \left[ \frac{n_q\cos\theta - n_T\sqrt{1 - \left(\frac{n_q}{n_T}\sin\theta\right)^2}}{n_q\cos\theta + n_T\sqrt{1 - \left(\frac{n_q}{n_T}\sin\theta\right)^2}} \right]^2 + \left[ \frac{n_q\sqrt{1 - \left(\frac{n_q}{n_T}\sin\theta\right)^2} - n_T\cos\theta}{n_q\sqrt{1 - \left(\frac{n_q}{n_T}\sin\theta\right)^2} + n_T\cos\theta} \right]^2 \right\} \right\}$$

(3)

where $n_q$ and $n_T$ are refractive indexes of quartz and TiO₂,

respectively. As demonstrated in Eq. (3), $E_i$ and $\theta$ remained the same in the two media as mentioned before. $n_q$ and $n_T$ are inherent properties of quartz and TiO₂, respectively, which also did not change. Thus, $E_{R,dis}$ remains constant regardless of the external medium being air or water. Therefore, the variable nature of $E_{E,dis}$ and constant feature of $E_{R,dis}$ cause differences in $E_{dis}$ and thus verify the existence of evanescent waves in TiO₂-QOFs. The existence of evanescent waves was also proven by tracking the irradiance loss in a UV irradiated uncoated fiber immersed in methylene blue solutions (Supplementary Note 5B).

**Modeling light propagation in TiO₂-QOFs**. We developed an energy balance model to simulate the light propagation along TiO₂-QOFs using geometrical optics in accordance with our schematics of TiO₂ coating layers on the quartz optical fibers and the 3 modes of light propagation. In this model, light rays at different angles are emitted from an LED light source and launched into the TiO₂-QOFs. After entering TiO₂-QOFs, each ray strikes the inner surface of TiO₂-QOFs with $\theta$ between $0.376\pi$ and $0.495\pi$. The model assumes $\theta$ is discrete with an increment of $0.0001\pi$, and each light ray has the same amount of radiant energy ($E_0$). Each light ray continuously strikes the inner TiO₂-QOF surfaces, which (i) generates evanescent waves at the quartz/water/TiO₂ interface and activates TiO₂ and/or returns into the fiber, or (ii) activates attached TiO₂ through refraction at the

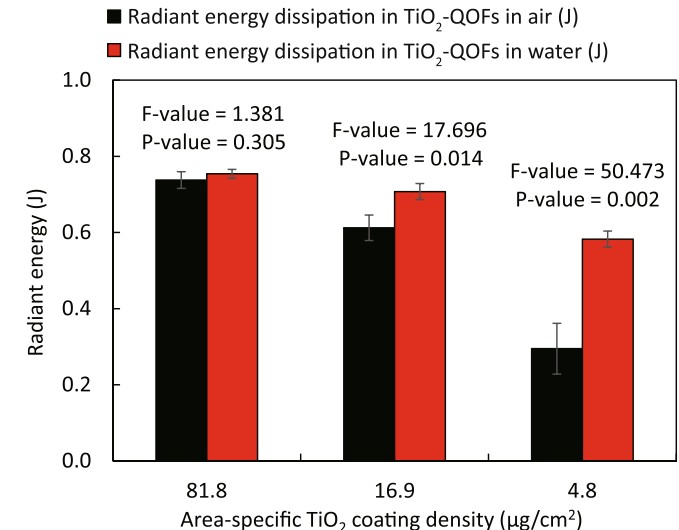

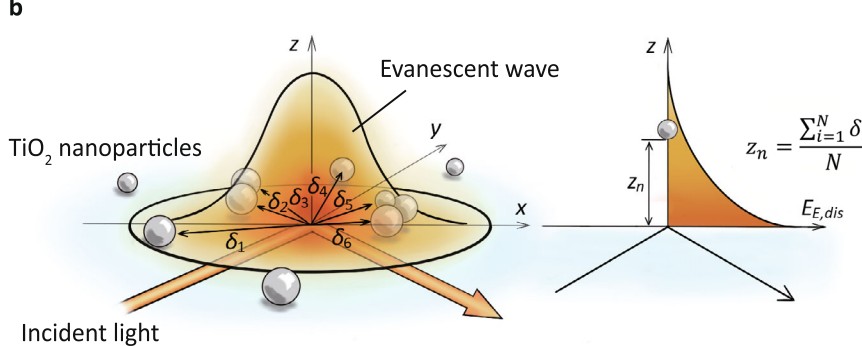

**Fig. 3 Radiant energy dissipated in TiO2-QOFs. a** Radiant energy dissipated in the three TiO2-QOFs when TiO2-QOFs were exposed to air and water; **b** schematic of the normalization of the distances between the optical fiber surface and the coated TiO2 nanoparticles in an evanescent field (conditions: light intensity = 7.02 mW/cm², wavelength = 365 nm, TiO2 coating length = 6.5 cm, irradiation duration = 4 h).

quartz/TiO2 interface with a small portion reflected to the fiber. Based on the above assumptions and Eqs. (1–3), the dissipated radiant energy of evanescent waves ($E_{E,dis}'$) and that of refracted light ($E_{R,dis}'$) of each light ray are represented by the summation of radiant energy dissipated at each TIR spot and refraction spot along TiO2-QOFs, respectively. The deterministic form of the equations is shown in Eqs. (4–6) (see Supplementary Note 5C for detailed derivation).

$$E_{E,dis}' = E_0 \frac{(1-p)e^{-\frac{z_a}{\Lambda}}\left\{1 - \left[(1-p)\left(1-e^{-\frac{z_a}{\Lambda}}\right) + p(1-T)\right]^{L/(d\tan\theta)}\right\}}{(1-p)\cdot e^{-\frac{z_a}{\Lambda}} + p\cdot T} \quad (4)$$

$$E_{R,dis}' = E_0 \frac{pT\left\{1 - \left[(1-p)\left(1-e^{-\frac{z_a}{\Lambda}}\right) + p(1-T)\right]^{L/(d\tan\theta)}\right\}}{(1-p)e^{-\frac{z_a}{\Lambda}} + pT} \quad (5)$$

where $p$ is the TiO2 patchiness on quartz optical fibers (Fig. 4a), $z_a$ is the average interspace distance between the optical fiber surface and the coated TiO2 layer (nm), which equals to the average value of $z_n$ at all TIR spots along the fiber length (Fig. 4b), $L$ is the TiO2 coating length (cm), and $d$ is the diameter of optical fibers (cm). $E_{dis}$ equals to the summation of $E_{E,dis}'$ and $E_{R,dis}'$ at all incident rays with $\theta$ from $0.376\pi$ to $0.495\pi$ as shown in Eq. (6).

$$E_{dis} = \sum_{\theta=0.376\pi}^{0.495\pi} E_{E,dis}' + \sum_{\theta=0.376\pi}^{0.495\pi} E_{R,dis}' \quad (6)$$

For our aqueous pollutant degradation study, an LED was attached to a single quartz optical fiber centered in the axial (i.e., longitudinal direction) of a tubular reactor filled with water containing carbamazepine. $E_{dis}$ is a function of $E_0$, $L$, $d$, and $n_e$, while $p$ and $z_a$ are intrinsic properties of the TiO2 coating layer and relate only to the coating structure but not the experimental conditions. $p$ and $z_a$ for each TiO2-QOF can then be calculated by substituting $E_0$, $L$, $d$, $n_e$, and $E_{dis}$ in air or water obtained by optical measurements when a TiO2-QOF is surrounded by air or immersed in water into Eqs. (4–6). Three experimental conditions were used to calculate the average $p$ and $z_a$ of each TiO2-QOF (detailed calculations for each condition shown in Supplementary Note 5D). The results showed $p$ and $z_a$ under the three conditions varied by <5% (Supplementary Table 2). The model's accuracy was also validated by comparing the measured $E_{dis}$ with the model-predicted values for incident light irradiance from 0.41 to 7.02 mW/cm² (details in Supplementary Note 5E).

The average $p$ and $z_a$ for TiO2-QOF-High were 0.528 cm²/cm² and 7.7 nm, respectively. For TiO2-QOF-Med, $p$ and $z_a$ were 0.206 cm²/cm² and 52.9 nm, respectively, and for TiO2-QOF-Low, they were 0.034 cm²/cm² and 114.3 nm, respectively (shown

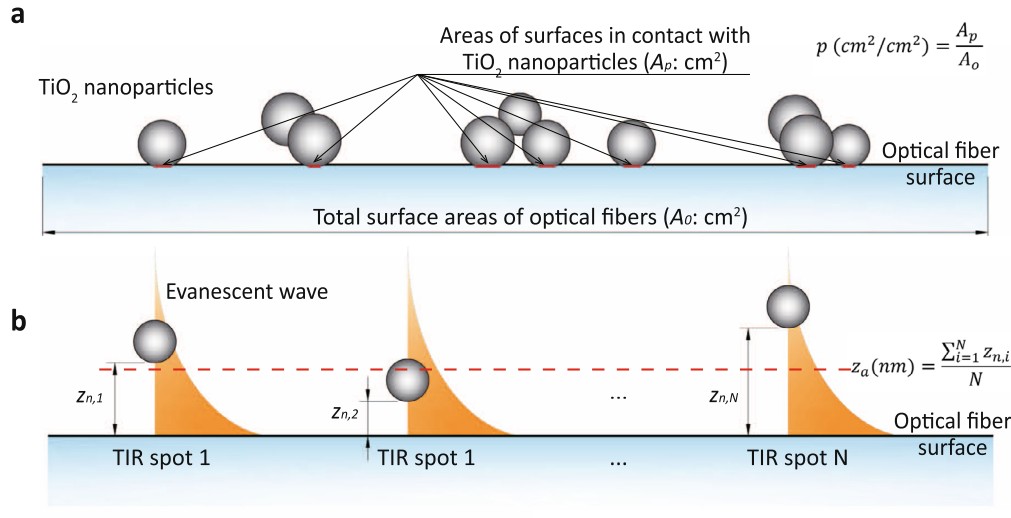

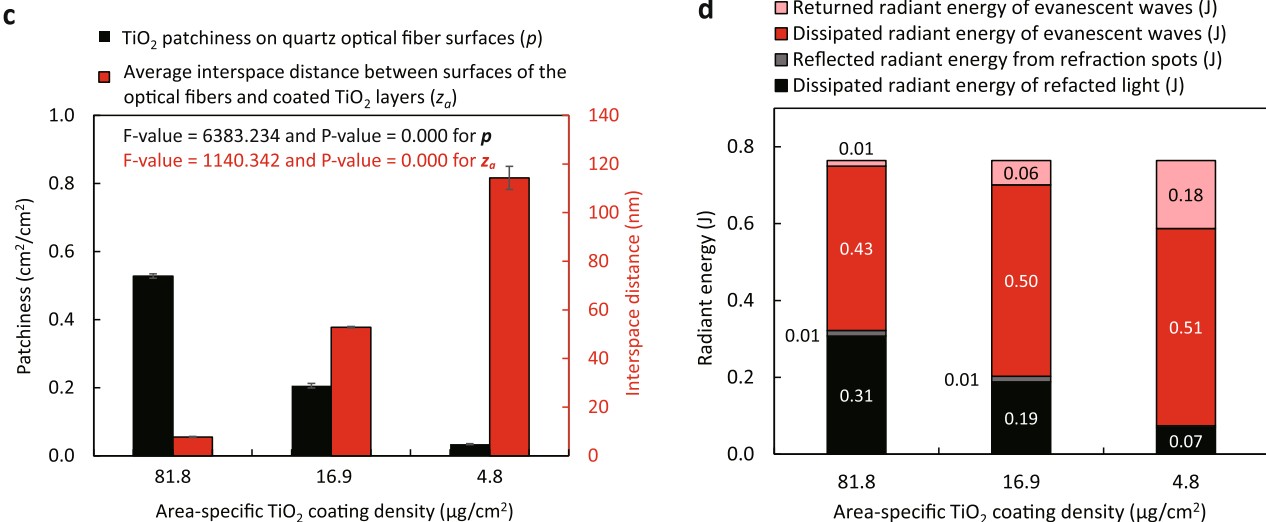

**Fig. 4 Modeling results of light propagation in the three TiO₂-QOFs. a** Schematic showing TiO₂ patchiness on quartz optical fibers ($p$); **b** schematic showing average interspace distance between fiber surfaces and TiO₂ coating layers ($z_a$); **c** TiO₂ coating layer structure parameters ($p$ and $z_a$) of the three TiO₂-QOFs; and **d** modeling results of the dissipated radiant energy of evanescent waves ($E_{E,dis}'$) and refracted light ($E_{R,dis}'$) as well as the returned radiant energy of evanescent waves ($E_{E,return}$) and the reflected radiant energy from refraction spots ($E_{R,reflect}$) in the three TiO₂-QOFs (conditions: light intensity = 7.02 mW/cm², wavelength = 365 nm, TiO₂ coating length = 6.5 cm, irradiation duration = 4 h).

in Fig. 4c). These results were consistent with the TiO₂ coating structure parameters we observed in the TEM images shown in Fig. 2. Within the same TiO₂-QOF, similar $p$ and $z_a$ calculated from the model under the three conditions further confirmed they only related to the coating structures rather than to experimental conditions.

Based on the average $p$ and $z_a$ obtained above, $E_{E,dis}'$ and $E_{R,dis}'$ can be calculated using Eqs. (4) and (5). In addition, to close the energy balance, the returned radiant energy of evanescent waves ($E_{E,return}$) and the reflected radiant energy from refraction spots ($E_{R,reflect}$) were calculated using Eqs. (7) and (8) (see Supplementary Note 5F for derivation).

$$E_{E,return} = \sum_{\theta=0.376\pi}^{0.495\pi}\left(E_0 - E_{E,dis}' - E_{R,dis}'\right)\frac{(1-p)\left(1-e^{-\frac{z_a}{\Lambda}}\right)}{p(1-T)+(1-p)\left(1-e^{-\frac{z_a}{\Lambda}}\right)} \quad (7)$$

$$E_{R,reflect} = \sum_{\theta=0.376\pi}^{0.495\pi}\left(E_0 - E_{E,dis}' - E_{R,dis}'\right)\frac{p(1-T)}{p(1-T)+(1-p)\left(1-e^{-\frac{z_a}{\Lambda}}\right)} \quad (8)$$

Fig. 4d compares the calculated $E_{E,dis}'$, $E_{R,dis}'$, $E_{E,return}$, and $E_{R,reflect}$ in water against area-specific TiO₂ coating densities. As the

area-specific TiO₂ coating density decreased from TiO₂-QOF-High to TiO₂-QOF-Med to TiO₂-QOF-Low, the generated evanescent wave energy ($E_{E,g}$)—i.e., the sum of $E_{E,dis}'$ and $E_{E,return}$,—increased from 0.44 to 0.56 to 0.69J, and the generated refracted light energy ($E_{R,g}$)—i.e., the sum of $E_{R,dis}'$ and $E_{R,reflect}$,—decreased from 0.32 to 0.20 to 0.07J. The increasing $E_{E,g}$ and decreasing $E_{R,g}$ from TiO₂-QOF-High to TiO₂-QOF-Med and then to TiO₂-QOF-Low confirmed that TiO₂-QOF-Low, which has the lowest TiO₂ patchiness, promotes Mode 3 and generates more evanescent waves.

We further compared the ratios of $E_{R,dis}'$ to $E_{R,g}$ and those of $E_{E,dis}'$ to $E_{E,g}$ in the three TiO₂-QOFs. All ratios of $E_{R,dis}'$ to $E_{R,g}$ were close to 1, confirming that refracted light mostly propagated away from TiO₂-QOFs and its radiant energy dissipated out. However, with increasing $E_{E,g}$ from TiO₂-QOF-High to TiO₂-QOF-Low, the ratios of $E_{E,dis}'$ to $E_{E,g}$ decreased from 0.98 to 0.74. This is because $z_a$ of TiO₂-QOFs modulates the evanescent wave energy returning to optical fibers. The value of $z_a$ in TiO₂-QOF-High (7.7 nm) is much smaller than the value of $\Lambda$ from quartz to water (50–120 nm) (Supplementary Fig. 11), suggesting

evanescent wave energy dissipates when they reach the closest $TiO_2$ nanoparticles and a negligible amount of them returned to the optical fibers to give a highest ratio of $E_{E,dis}'$ to $E_{E,g}$. As increasing $z_a$ from 7.7 nm in $TiO_2$-QOF-High to 52.9 nm in $TiO_2$-QOF-Med and 114.3 nm in $TiO_2$-QOF-Low, more evanescent waves returned to the optical fibers (Supplementary Fig. 12) because they cannot reach the $TiO_2$ coating layers to give lower ratios of $E_{E,dis}'$ to $E_{E,g}$. Those trends were also observed when the $TiO_2$-QOFs were exposed to air (Supplementary Fig. 13). The above modeling results suggest the $TiO_2$ layer structure parameters ($p$ and $z_a$) are important to control the light energy within $TiO_2$-QOFs, while the $TiO_2$ coating thickness is not critical.

**Degradation of carbamazepine by $TiO_2$-QOFs.** The above data, models, and insights were based on light measurements outside and along the length of fibers coated with $TiO_2$. We then examined the impact of increasing energy efficiency of the low patchiness $TiO_2$ coating layer to its photocatalytic performance in carbamazepine degradation. Control tests confirmed that carbamazepine was not adsorbed by $TiO_2$-QOFs in the dark or not degraded when light was launched from LEDs into an uncoated optical fiber (Supplementary Figs. 14a and b). Even though the radiant energy dissipation varied in the three $TiO_2$-QOFs, launching 365 nm light from an LED separately into the three $TiO_2$-QOFs resulted in statistically the same ($p = 0.450$) pseudo-first-order degradation kinetics for carbamazepine in water (rate constants $k$, shown in Fig. 5a). The calculated quantum yields of carbamazepine degradation (i.e., moles of carbamazepine degraded per mole of photons absorbed by $TiO_2$ coating layers) increased from 0.0189 in $TiO_2$-QOF-High to 0.0248 in $TiO_2$-QOF-Low (Fig. 5b). The improved quantum yield was attributed to the highest quantity of evanescent waves generated in $TiO_2$-QOF-Low saving more radiant energy than the other two. The evanescent waves generated in $TiO_2$-QOF-Low allow light to be evenly dissipated along the fiber, which prevents local photon oversaturation at the beginning sections of $TiO_2$-QOFs and thus reduces its associated efficiency losses (Supplementary Note 6A). We also proved that the improved quantum yield was not attributed to the mass transfer limitation (Supplementary Note 6B) or carbamazepine adsorption (Supplementary Fig. 14c) of different $TiO_2$ coating layers. We then fabricated two new $TiO_2$-QOFs, i.e., $TiO_2$-QOF-Low'' at $p$ of 0.018 and $z_a$ of 139.50 nm and $TiO_2$-QOF-Low' at $p$ of 0.026 and $z_a$ of 127.97 nm. Both degradation rate constants and quantum yields decreased with decreasing $p$ and increasing $z_a$ (Supplementary Table 5).

Quantum yields of carbamazepine degradation by the $TiO_2$-QOF-Low are 1–50× higher than those in the UV-based advanced oxidation processes[34–36] (Supplementary Table 6). The quantum yield of $TiO_2$-QOF-Low can be further increased by extending its length to fully utilize the returned radiant energy of evanescent waves in $TiO_2$-QOF-Low and to increase its photocatalytic reactive sites. Figure 5c shows the percentage of radiant energy dissipation relative to total radiant energy delivered into $TiO_2$-QOFs with different coatings. Denser coatings on $TiO_2$-QOF-High resulted in >92% of the energy launched into the fiber being dissipated within the first 6.5 cm of the coated length, whereas low patchiness $TiO_2$-QOF-Low utilized the same amount light over 26 cm of the coated length. The 4× increase in length also increased by 4× the surface areas available for ROS production that degrades pollutants, therefore, the 26-cm $TiO_2$-QOF-Low was estimated to achieve 2× degradation rate constants and 2× apparent quantum yields (moles of carbamazepine degraded per moles of photons launched to optical fibers) compared with $TiO_2$-QOF-High at a coating length of 6.5 cm (Supplementary Note 6D). The higher rate constant and apparent quantum yield of 26-cm $TiO_2$-QOF-Low was further experimentally confirmed in a 70 mL reactor (Fig. 5d). Even if refracted light emitted from $TiO_2$-QOF-High can be utilized by bundling $TiO_2$-QOFs together[37], optimizing evanescent waves to fully use transmitted light in longer $TiO_2$-QOF-Low is 44–96% more efficient (in carbamazepine degradation rates and apparent quantum yields) than harvesting refracted light by using $TiO_2$-QOF-High bundles (detailed see Supplementary Note 6E). By controlling surface patchiness and distance between fiber surface and photocatalyst coating layers, $TiO_2$-QOF-Low not only prevent light oversaturation and its associated efficient losses, but also reduces light wasted by refraction and increase surface reactive sites. These features make $TiO_2$-QOF-Low more energy-efficient to degrade pollutants. In addition, the 26-cm $TiO_2$-QOF-Low used 77% fewer $TiO_2$ than the 6.5-cm $TiO_2$-QOF-High, although the former was 4× longer, and it had more surface areas available for photocatalytic reactions. The $TiO_2$ coating in $TiO_2$-QOF-Low is also stable, and there was no $TiO_2$ coming off during the reaction (Supplementary Fig. 19).

**Implications for photocatalytic reactor design.** Photocatalytic processes have broad application in water and air treatment, energy production, organic synthesis, and other fields. Less than 1 in 100 papers address the critical barrier to making photocatalytic reactors more effective, namely light and energy management; the other papers focus largely on discovery of new or incremental improvement in existing photocatalyst materials[10]. An important but often overlooked aspect to limiting use of photocatalytic processes into engineering practice is the influence of photocatalytic reactor design, rather than material properties alone. To address the barrier of reactor design rather than material properties we show how quantum yields are effectively increased more significantly by managing the way light reaches the catalyst, than recent improvements in catalyst materials themselves. In addition to considering only refracted light in our optical fiber reactor, another key insight was the separation of photocatalyst activation by refracted light versus evanescent wave energy to activate the photocatalyst. We report for the first time here the relative importance of both mechanisms, as well as how to modulate their relative importance. Here, we developed a novel coating strategy that left 3.4% of the optical fiber surface in direct contact with $TiO_2$ and, on-average, 114.3 nm interspace distance between the fiber surface and the coated $TiO_2$ layers. This strategy successfully reduced refraction, generated evanescent waves in the $TiO_2$-QOF, and thus allowed even dissipation of light along $TiO_2$-QOF to prevent oversaturation of the light delivered to the fiber and its associated efficiency losses. The novel $TiO_2$-QOF enables the design of a photocatalytic reactor that uses 77% less mass of photocatalysts but achieves up to 96% improvement in quantum yields compared with reactors built of fibers with densely coated $TiO_2$ layers, mainly because optimizing evanescent waves to fully use transmitted light in longer $TiO_2$-QOF-Low is more efficient than harvesting refracted light by using $TiO_2$-QOF-High bundles. Moreover, such a strategy of managing evanescent wave energy is applicable to all photocatalysts and thus emerges as an important cost consideration especially when contemplating expensive photocatalysts.

A new platform can also be established using novel photocatalyst-coated optical fibers to allow evanescent wave energy to activate photocatalysts. This new platform can quantify performance of photocatalysts more precisely by minimizing the radiation scattering by photocatalysts and water parameters, preventing aggregation of photocatalysts to maximize the interaction between reactive sites and target compounds, and allowing easier and more accurate quantification of photons absorbed by photocatalysts.

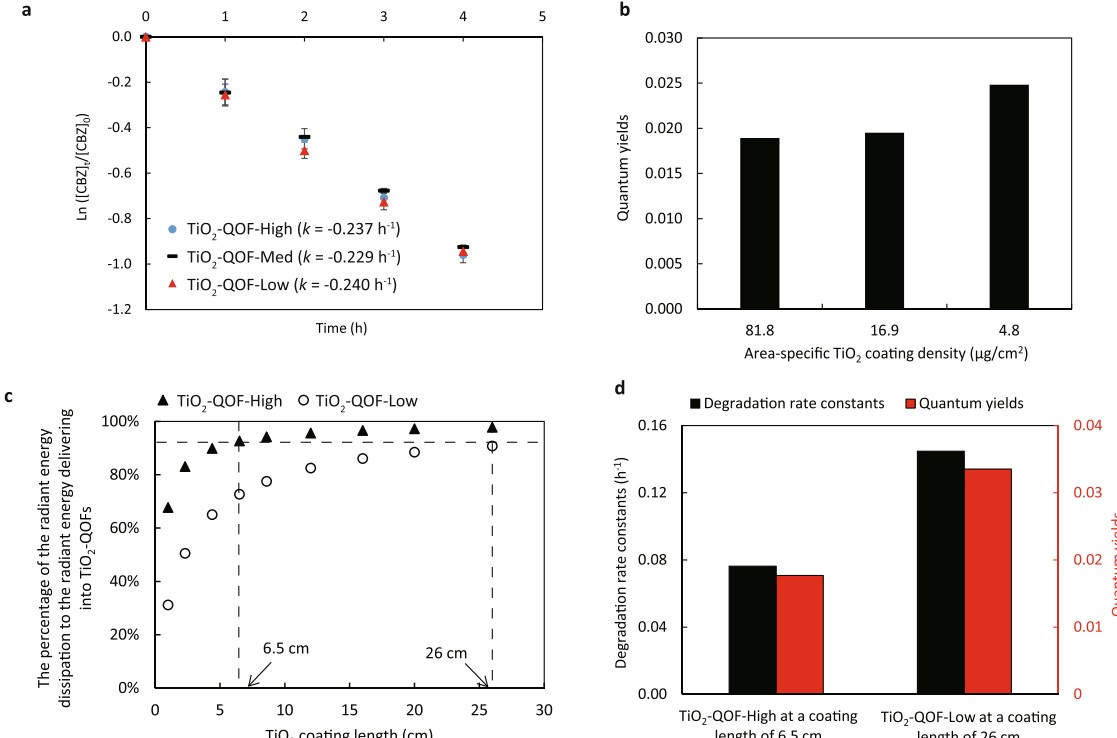

**Fig. 5 Carbamazepine degradation. a** Pseudo-first order degradation kinetics; **b** quantum yields of carbamazepine degradation by the three TiO₂-QOFs; **c** the percentage of the radiant energy dissipation to the radiant energy delivered to TiO₂-QOFs as a function of TiO₂ coating length; and **d** comparison between degradation rate constants and apparent quantum yields by the UV irradiated TiO₂-QOF-High at a coating length of 6.5 cm and TiO₂-QOF-Low at a coating length of 26 cm in a 70 mL reactor (conditions: light intensity = 7.02 mW/cm², light wavelength = 365 nm, [CBZ]₀ = 2 μM, irradiation duration = 4 h).

## Methods

**Coating quartz optical fibers with TiO₂ layers**. Uncoated optical fibers (FT1000UMT, Thorlabs) were prepared by cutting the fibers into segments of specified lengths, stripping the buffer coating and cladding, and polishing both tips (described in Supplementary Method A). TiO₂ suspension was prepared by dispersing TiO₂ (P25) in double deionized water (18.2 MΩ-cm). TiO₂-QOFs were fabricated by dipping the optical fiber segments in TiO₂ suspension at different conditions (Table 1) to produce TiO₂ layers with different area-specific TiO₂ coating densities[20]. TiO₂-QOF-High was fabricated by completing 5 cycles of dipping the segments in a 10,000 mg/L TiO₂ suspension for 0.5 min and air drying for another 0.5 min. TiO₂-QOF-Med was fabricated following a similar method, but with only one dip-coating/drying cycle. TiO₂-QOF-Low was fabricated by dipping the segments in a 40 mg/L TiO₂ suspension for 1 h followed by air drying.

**Characterization of TiO₂-QOFs**. The images of coated and uncoated fiber surfaces were obtained by a SEM (JSM-6700F, JEOL) and an AFM (Dimension 3100, Digital Instruments). The cross-sections were prepared by a FIB following a standard preparation method[38] (details in Supplementary Method B), before characterization by a TEM (JEM-100CXII, JEOL). The cross-sections were also characterized and confirmed by a 3D surface optical profilometer (Wyko NT 3300, Veeco). The TiO₂ layer masses ($m_{TiO2}$) on the optical fibers were measured gravimetrically by the weight of the optical fibers before and after the dip-coating/drying cycles. The porosity of the TiO₂ coating layers, which is defined as the fraction of the total pore volume over the volume of the TiO₂ layer, was calculated using Eq. (9).

$$Porosity = 1 - \frac{m_{TiO2}}{\rho_{TiO2}L\pi dD} \tag{9}$$

where $\rho_{TiO2}$ is the true density of the TiO₂ particles (4.26 g/mL at 25 °C obtained from Sigma-Aldrich), $L$ is the TiO₂ coating length, $d$ is the diameter of optical fibers, and $D$ is the thickness of the TiO₂ coating layers, which was determined by the cross-sectional profiles of TiO₂-QOFs from SEM images. The TiO₂ patchiness was calculated by dividing the optical fiber surface in direct contact with TiO₂ nanoparticles by the total surface of the optical fiber determining from TEM cross-section images.

**Photocatalytic experiment**. The photocatalytic experiment including irradiance measurement and carbamazepine degradation was conducted in a mixed batch reactor as shown in Supplementary Fig. 7. The reactor was composed of a cylindrical glass vessel (23 mL) with a length of 65 mm and an inner diameter of

25 mm, a magnetic stirrer (F203A0160, VELP) at the bottom for rapid mixing, a 365 nm LED light source (H44LV1C0, HPLighting), and an optical meter (RPS900-R, International Light Technologies). A single TiO₂-QOF was fixed in the reactor with one end mounted to the LED and the other end connected to the optical meter to obtain the transmitted irradiance after the 6.5 cm light path. All the experiments were conducted in triplicate, and TiO₂-QOFs were fabricated using the same preparation method in every repeated experiment. The radiant energy dissipated in the TiO₂-QOFs ($E_{dis}$: J) was expressed as the difference between radiant energy delivered into TiO₂-QOFs ($E_{in}$) and the radiant energy transmitted to the terminal end ($E_{out}$). The radiant energy of the uncoated optical fiber measured at the terminal end in air was regarded as $E_{in}$, because light delivered into an uncoated optical fiber is totally reflected, and attenuation along the fiber is negligible in a very short propagation distance. $E_{dis}$ is expressed as Eq. (10).

$$E_{dis} = (I_{in} - I_{out})At \tag{10}$$

where $I_i$ is the irradiance measured from the optical meter (W/cm²), $A$ is the cross-sectional area of optical fibers (cm²), and $t$ is the irradiation duration (s). Carbamazepine concentrations were determined using a high-performance liquid chromatograph (VP series, Shimadzu) equipped with a Waters symmetry C18 column and a UV-Vis detector. The degradation kinetic was obtained by plotting carbamazepine concentration as a function of time to obtain the reaction order and rate constants ($k$: h⁻¹). The quantum yields ($\eta$) of the degradation was calculated using Eq. (11).

$$\eta = \frac{kV([CBZ]_0)^1}{I_{dis}A} \tag{11}$$

where $[CBZ]_0$ is the initial carbamazepine concentration (mole/L), $V$ is the liquid volume (L), and $I_{dis}$ is the irradiance dissipated in the TiO₂-QOFs (mol-photons/(cm²·h)).

## Data availability

The authors declare that the data that support the findings of this study are available from the corresponding author upon reasonable request.

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

## Acknowledgements

This work was partially funded by the Hong Kong Research Grants Council (16202219 and T21-604/19-R) and the National Science Foundation (EEC-1449500) Nanosystems Engineering Research Center on Nanotechnology-Enabled Water Treatment.

## Author contributions

Y.S. and L.L. involve in the processes of conceived and designed the analysis, collected the data, contributed data or analysis tools, performed the analysis, and wrote the paper. P.W. and C.S. involve in the processes of performed the analysis and wrote the paper.

## Competing interests

The authors declare no competing interests.
