## [Peer Review File · Nature Communications]

REVIEWER COMMENTS

Reviewer #1 (Remarks to the Author):

The present manuscript describes a new approach for photocatalyst-coated optical fibers. Instead of mainly relying on refraction to transfer the light into the photocatalyst, this is achieved by evanescent waves. The authors argue that refraction has the disadvantage that either, when using thin coatings, a large fraction of the light is transmitted through the coating or when using thicker coatings, mass transport limitation may be a problem. Indeed, this „backside“ illumination of thicker porous layers would create opposing gradients of absorbed light flux and pollutant diffusion, which are not ideal and can lead to mass-transfer limitations. As such the paper is interesting and presents an intriguing new way of performing photocatalysis.

However, I am skeptical about the proposed advantages and claimed higher energy efficiency. My assessment on whether this paper has impact sufficiently high for publication in Nature Communications is dependent on whether the authors can prove their system really is more efficient on a fair basis (vide infra). Therefore I recommend a major revision based on the following aspects.

The author's universal claim of a higher energy efficiency of evanescent wave energy transfer versus refraction is not proven for the basic case of 6.5 cm fiber length. In this case, the observed degradation rates are almost exactly equal. The quantum yield of the low coating strength fiber may be the highest as less light is lost to refraction, but this is compensated by higher transmission losses - which are losses all the same.

So in this case, the claimed superiority is only achieved by defining refraction losses to be inside the frame of considered effects and transmission losses outside. However, quantum yield as it is usually defined is the ratio of reaction events to absorbed photons. In this definition, photons being refracted and transmitted through the TiO₂ coating (and are not absorbed) should not be considered, the same as photons transmitted through the fiber.

On the other hand, the apparent quantum yield considers the ratio of reaction events to photons supplied to the system, neglecting transmission and reflection losses. In this case, both photons transmitted through the coating and the fiber should be considered. In either case, the proposed higher quantum yield of the evanescent wave system is only achieved by an unconventional interpretation of said efficiencies.

Only in the case of the longer optical fiber the lower coating system does achieve a higher reaction rate and also a higher quantum yield. Therefore, a general claim of higher efficiency cannot be made but must be conditional in that to achieve the higher efficiency also longer fibers and larger reactors are required.

Moreover, the authors state that light that is refracted out of the fiber and not absorbed by the TiO₂

layer is “lost to water”. This may be true for UVC light but not for the present case of using UVA as water only has a (in this case) negligible absorption coefficient for this radiation. Moreover, in practice, not a single but multiple fibers will be used in a reactor. Light that is refracted out of the fiber into the medium will therefore in most cases not be lost but absorbed by the TiO₂ coating of another fiber. This is true at least as long as the reaction medium is somewhat transparent to UVA radiation and does not contain strongly UVA-absorbing solutes. As the author’s main claim is that superiority of evanescent wave energy transfer versus refraction, the model should also encompass a fair comparison with a bundle of fibers where refracted light may be absorbed by another fiber’s coating.

Minor points:

- The comparisons of fiber coating strength would be more appropriately given area-specifically, i.e. mg/cm² of fiber surface.
- The presence of evanescent waves was not explicitly proven, as claimed on several passages of the paper, merely inferred. The authors should be clear about this. I am not an expert on optics so I not aware if there are more explicit ways to prove this, if there are, the authors should try to generate explicit direct evidence. Alternatively, a second indirect experiment may help. For instance using uncoated fibers in a solution with a strongly UVA absorber. In this case there should also be evanescent wave energy transfer, and its intensity could be compared to the low coating fibers. If no additional evidence is provided, the authors should refrain from the claim of having proven evanescent energy transfer.

Reviewer #2 (Remarks to the Author):

In this paper, the authors proposed a strategy to maximize photocatalytic optical fiber light usage and quantified interactions between two forms of light energy (refracted light and evanescent waves) and surface-coated photocatalysts. They conclude that maximizing evanescent waves to activate photocatalysts by controlling photocatalyst coating structures emerged as an effective strategy to improve light usage in photocatalysis. However, in the photocatalytic reaction, not only the light absorption process but also multiple factors such as substrate adsorption, desorption, and redox reaction affect the photocatalytic activity. Currently, the author focuses only on the light absorption process, and the comprehensive evaluation is insufficient. It can be recommended to be considered for publication in this journal after major revision as suggested by following comments.

Some additional comments:

1. If the photocatalyst layer is thick, TiO₂ existing at the interface with the fiber absorbs light, but TiO₂ far from the fiber may not receive sufficient light. In that case, TiO₂-QOF-High, which is disadvantageous in terms of mass transfer, shows low activity. The author concludes that the ratio of refracted light to evanescent waves is a factor in the difference in activity, but it is necessary to exclude that the

difference in mass transfer affects the activity.

2. It is necessary to observe the leakage of refracted light from the each POF. If no refracted light can be observed, it means that TiO₂ layer is completely absorbed the leakage light and the light absorption process does not affect the difference in photocatalytic activity. Mass transfer and adsorption due to the difference in the thickness of the TiO₂ layer may affect the photocatalytic activity.

3. In photocatalytic decomposition experiments, the change in concentration of CBZ until the adsorption equilibrium is reached should be shown. If it does not adsorb, what is the mechanism by which decomposition is considered to be proceeding?

4. What is oxidized and reduced in the photocatalytic reaction? Should be shown with a band structure model of the prepared photocatalyst.

5. The photocatalytic decomposition data presented by the authors showed the highest activity in POF with the least photocatalytic coating. Is the thinner the coating of the photocatalyst layer, the higher the activity? Is there a peak in the relationship between photocatalytic amount and activity?

6. What solvent was used for dip coating of TiO₂? During the photocatalytic reaction, is the coating come off without calcination?

7. TiO₂-QOF-High, which has a large amount of deposition, should have more TiO₂ far from the optical fiber surface than other POFs. Why is α of TiO₂-QOF-Low the largest?

Title: Evanescent waves modulate energy efficiency of photocatalysis within nano-TiO₂ coated quartz optical fibers illuminated using LEDs

Authors: Yinghao Song, Li Ling, Paul Westerhoff, Chii Shang

Manuscript ID: NCOMMS-20-44004-T

Before responding to the comments of the reviewers, the authors would like to express our deepest appreciation for their time and effort in reviewing this manuscript.

Responses to comments of Reviewer 1:

The present manuscript describes a new approach for photocatalyst-coated optical fibers. Instead of mainly relying on refraction to transfer the light into the photocatalyst, this is achieved by evanescent waves. The authors argue that refraction has the disadvantage that either, when using thin coatings, a large fraction of the light is transmitted through the coating or when using thicker coatings, mass transport limitation may be a problem. Indeed, this "backside" illumination of thicker porous layers would create opposing gradients of absorbed light flux and pollutant diffusion, which are not ideal and can lead to mass-transfer limitations. As such the paper is interesting and presents an intriguing new way of performing photocatalysis.

However, I am skeptical about the proposed advantages and claimed higher energy efficiency. My assessment on whether this paper has impact sufficiently high for publication in Nature Communications is dependent on whether the authors can prove their system really is more efficient on a fair basis (*vide infra*). Therefore, I recommend a major revision based on the following aspects.

Comment 1:

The author's universal claim of a higher energy efficiency of evanescent wave energy transfer versus refraction is not proven for the basic case of 6.5 cm fiber length. In this case, the observed degradation rates are almost exactly equal. The quantum yield of the low coating strength fiber may be the highest as less light is lost to refraction, but this is compensated by higher transmission losses - which are losses all the same.

So in this case, the claimed superiority is only achieved by defining refraction losses to be inside the frame of considered effects and transmission losses outside. However, quantum yield as it is usually defined is the ratio of reaction events to absorbed photons. In this definition, photons being refracted and transmitted through the TiO₂ coating (and are not absorbed) should not be considered, the same as photons transmitted through the fiber.

On the other hand, the apparent quantum yield considers the ratio of reaction events to photons supplied to the system, neglecting transmission and reflection losses. In this case, both photons transmitted through the coating and the fiber should be considered. In either case, the proposed higher quantum yield of the evanescent wave system is only achieved by an unconventional interpretation of said efficiencies.

Only in the case of the longer optical fiber the lower coating system does achieve a higher reaction rate and also a higher quantum yield. Therefore, a general claim of higher efficiency cannot be made but must be conditional in that to achieve the higher efficiency also longer fibers and larger reactors are required.

Reply:

The reviewers make a good point that we need to more clearly articulate. We believe the comparison in energy efficiency between evanescent waves and refracted light is on an equivalent basis for two different fiber lengths (6.5 and 26 cm lengths). The light energy launched to TiO₂-QOFs equals to the sum of i) the light energy absorbed by TiO₂ (through evanescent waves and refracted light), ii) refraction losses (light energy transmitted through TiO₂ coatings), and iii) transmission losses (light energy transmitted through the optical fiber). The light irradiance emitted from the side of the TiO₂-QOFs accounts only for <1.35% of the irradiance launched to TiO₂-QOFs (detailed see Comment 2), suggesting light refracted from quartz fibers is absorbed by TiO₂ layers and refraction losses can be neglected in this study.

When assessing the quantum yield, which is defined as the ratio of reaction events to absorbed photons, the absorbed photonic energy comprises only the evanescent wave and refracted light energy absorbed by TiO₂. Since light refracted from quartz fibers is absorbed by TiO₂ layers, TiO₂-QOF-Low with high quantity of evanescent waves absorbed less light energy compared with TiO₂-QOF-High and TiO₂-QOF-Med. Therefore, TiO₂-QOF-Low showed high quantum yields for at a fiber length of 6.5 cm as shown in the original Manuscript Fig. 5(b).

When assessing the apparent quantum yield, which is defined as the ratio of reaction events to photons launched to the system, both the light energy absorbed by TiO₂ and the transmission losses should be considered. The apparent quantum yield of TiO₂-QOF-Low at 6.5 cm shows no superiority compared with TiO₂-QOF-High or TiO₂-QOF-Med. However, in Supplementary Section 6.6, we already simulated the carbamazepine degradation rate constants and apparent quantum yields of TiO₂-QOF-Low at a fiber length of 26 cm. Their values were both 2× higher than those of TiO₂-QOF-High at a coating length of 6.5 cm in a 92 mL reactor. This simulation was further confirmed by our additional experiments in response to the reviewer's comment (Fig. 5d), in which the 26 cm TiO₂-QOF-Low also showed 2× rate constants and 2× quantum yields than the 6.5 cm TiO₂-QOF-High when degrading carbamazepine in a 70 ml reactor.

These clarifications have been revised in **the revised Manuscript**:

Fig. 5d comparison between degradation rate constants and apparent quantum yields by the UV irradiated TiO₂-QOF-High at a coating length of 6.5 cm and TiO₂-QOF-Low at a coating length of 26 cm in a 70 mL reactor.

Lines 52–53, “(i.e., moles of pollutants degraded per mole of photons absorbed by photocatalysts)”,

Lines 273–274, “(i.e., moles of carbamazepine degraded per mole of photons absorbed by TiO₂ coating layers)”,

Lines 283–290, “When assessing the apparent quantum yield (moles of carbamazepine degraded per mole of photons launched to optical fibers), both the light energy absorbed by TiO₂ and the transmission losses should also be considered. The apparent quantum yield of TiO₂-QOF-Low at 6.5 cm shows no superiority compared with TiO₂-QOF-High or TiO₂-QOF-Med. However, its apparent quantum yield can be further increased by bundling multiple TiO₂-QOFs with a single LED as reported before or, as we show here, extending

the length of TiO₂-QOF-Low to utilize the returned radiant energy and increase the photocatalytic reactive sites.”

Lines 295–299, “therefore, the 26-cm TiO₂-QOF-Low was estimated to achieve 2× degradation rate constants and 2× apparent quantum yields compared with TiO₂-QOF-High at a coating length of 6.5 cm (Supplementary Section 6.6). The higher rate constant and apparent quantum yield of 26-cm TiO₂-QOF-Low was further experimentally confirmed in a 70 mL reactor (Fig. 5d). ”

Figure 5d. Comparison between carbamazepine degradation rate constants and apparent quantum yields (mole of pollutant degraded by mole of light launched to the fiber) by UV irradiated TiO₂-QOF-High at a coating length of 6.5 cm and TiO₂-QOF-Low at a coating length of 26 cm in a 70 mL reactor. (Conditions: light intensity = 7.02 mW/cm², wavelength = 365 nm, irradiation duration = 4 h)

Comment 2:

Moreover, the authors state that light that is refracted out of the fiber and not absorbed by the TiO₂ layer is “lost to water”. This may be true for UVC light but not for the present case of using UVA as water only has a (in this case) negligible absorption coefficient for this radiation. Moreover, in practice, not a single but multiple fibers will be used in a reactor. Light that is refracted out of the fiber into the medium will therefore in most cases not be lost but absorbed by the TiO₂ coating of another fiber. This is true at least as long as the reaction medium is somewhat transparent to UVA radiation and does not contain strongly UVA-absorbing solutes. As the author’s main claim is that superiority of evanescent wave energy transfer versus refraction, the model should also encompass a fair

comparison with a bundle of fibers where refracted light may be absorbed by another fiber's coating.

Reply:

The reviewer's comment is well taken regarding our comparison between evanescent waves and refracted light, and whether evanescent waves are really more energy efficient than refracted light. To quantify these energies, we measured the light irradiance 1 mm away from side surfaces of the three TiO₂-QOFs, as shown in Supplementary Table 4. The light irradiance emitted from side surface of all the three TiO₂-QOFs at 1 cm and 5 cm along the fiber length is less than 0.10 mW/cm² and 0.01 mW/cm², respectively, which account only for <1.35% and <0.15% of the irradiance launched to TiO₂-QOFs, respectively. Although more light energy was absorbed by TiO₂-QOF-High, 80% of the light energy launched to TiO₂-QOF-High was absorbed at the beginning 2 cm of the fiber, while only about 12% of the light energy was absorbed at the following 4.5 cm. Therefore, the carbamazepine degradation occurred mostly at the beginning 2 cm of the fiber, which limits the photocatalytic activity of TiO₂-QOF-High. On the other hand, the light energy distribution along TiO₂-QOF-Low was more uniform as shown in the original Supplementary Fig. 15. More TiO₂ reactive sites was used for photocatalytic degradation with less light energy. Therefore, TiO₂-QOF-Low showed a higher quantum yield.

Supplementary Table 4. Light refracted from side surfaces of TiO₂-QOFs

Name	Light launched to TiO ₂ -QOFs (mW/cm ²)	Light refracted at 1 cm of the TiO ₂ -QOFs (mW/cm ²)	Light refracted at 5 cm of the TiO ₂ -QOFs (mW/cm ²)
TiO ₂ -QOF-High	7.02	0.0942	0.0074
TiO ₂ -QOF-Med	7.02	0.0823	0.0102
TiO ₂ -QOF-Low	7.02	0.0339	0.0099

This result is added to **the revised Manuscript lines 247–248**: “This dissipated radiant energy was mostly absorbed by the TiO₂ coating layers (details see Supplementary Table 4).” This table is also added to **the revised Supplementary Information Section 5.7**.

The light irradiance emitted from the side of the three TiO₂-QOFs is too low to generate sufficient radicals on TiO₂ coatings of another fiber even when multiple TiO₂-QOFs are used. Moreover, in practice, aqueous components, such as NOM in water, absorb or scatter the light emitted from the side of TiO₂-QOFs, further inhibited light reaching TiO₂ coatings of other fibers. These results suggest that the light emitted from the side of TiO₂-QOFs cannot be effectively utilized by the TiO₂ coatings of another fibers in this study.

Comment 3:

The comparisons of fiber coating strength would be more appropriately given area-specifically, i.e. mg/cm² of fiber surface.

Reply:

The linear TiO₂ mass coating density (mg/cm) has been revised to the area-specific TiO₂ coating density (mg/cm²) accordingly in the revised manuscript.

Comment 4:

The presence of evanescent waves was not explicitly proven, as claimed on several passages of the paper, merely inferred. The authors should be clear about this. I am not an expert on optics so I not aware if there are more explicit ways to prove this, if there are, the authors should try to generate explicit direct evidence. Alternatively, a second indirect experiment may help. For instance, using uncoated fibers in a solution with a strongly UVA absorber. In this case there should also be evanescent wave energy transfer, and its intensity could be compared to the low coating fibers. If no additional evidence is provided, the authors should refrain from the claim of having proven evanescent energy transfer.

Reply:

We appreciate the suggestion, and performed additional experiments. The existence of evanescent waves on the interface between quartz and air/water has already been proven by previous studies¹. The presence of evanescent waves in the current manuscript was demonstrated by tracking the irradiance loss in a UV irradiated uncoated fiber immersed in methylene blue (MB) solutions as shown in Supplementary Fig. 9. When launching UV light to the uncoated fiber, the irradiance loss increased by around 10 times with increasing MB concentrations from 1.3 to 65 mg/L. The irradiance loss was due to the absorption of evanescent waves generated on uncoated quartz fiber surfaces by MB. This test supports the presence of evanescent waves.

Supplementary Figure 9. Evanescent waves generated at uncoated optical fiber surface absorbed by methylene blue as a function of methylene blue concentrations. (Conditions: light intensity = 17.6 mW/cm², wavelength = 365 nm)

This information is added to **the revised Manuscript:**

Lines 186–187: “The existence of evanescent waves was also proven by tracking the irradiance loss in a UV irradiated uncoated fiber immersed in methylene blue solutions (Supplementary Fig. 9).”

The reply is also added to **the revised Supplementary Information:**

Section 5.2: “The existence of evanescent waves on quartz optical fiber surfaces was demonstrated by tracking the irradiance loss in a UV irradiated uncoated fiber immersed in methylene blue (MB) solutions as shown in Supplementary Fig. 9. When launching UV light to the uncoated optical fiber, the irradiance loss increased by around 10 times with increasing MB concentrations from 1.3 to 64.5 mg/L. The irradiance loss was due to the absorption of evanescent waves generated on uncoated quartz fiber surfaces by MB. This test supports the presence of evanescent waves.”

Reference:

[1] Dieing, T., Hollricher, O. & Toporski, J. *Evanescent Wave in Optics*. (Springer Series in Optical Sciences, 2010)

Responses to comments of Reviewer 2:

In this paper, the authors proposed a strategy to maximize photocatalytic optical fiber light usage and quantified interactions between two forms of light energy (refracted light and evanescent waves) and surface-coated photocatalysts. They conclude that maximizing evanescent waves to activate photocatalysts by controlling photocatalyst coating structures emerged as an effective strategy to improve light usage in photocatalysis. However, in the photocatalytic reaction, not only the light absorption process but also multiple factors such as substrate adsorption, desorption, and redox reaction affect the photocatalytic activity. Currently, the author focuses only on the light absorption process, and the comprehensive evaluation is insufficient. It can be recommended to be considered for publication in this journal after major revision as suggested by following comments.

Comment 1:

If the photocatalyst layer is thick, TiO₂ existing at the interface with the fiber absorbs light, but TiO₂ far from the fiber may not receive sufficient light. In that case, TiO₂-QOF-High, which is disadvantageous in terms of mass transfer, shows low activity. The author concludes that the ratio of refracted light to evanescent waves is a factor in the difference in activity, but it is necessary to exclude that the difference in mass transfer affects the activity.

Reply:

We appreciate the comment. In response to the reviewer's comment, we evaluated the impacts of external mass transfer and internal mass transfer of the TiO₂ coating layers on the photocatalytic performance of different TiO₂-QOFs. The external mass transfer describes the diffusion of pollutants from bulk solutions to the TiO₂ coating surface. It is a function of the Reynolds number¹. Therefore, external mass transfer of the three TiO₂-QOFs are the same in the current manuscript because the reactor setup is the same. On the other hand, the internal mass transfer describes the diffusion of pollutants and radicals inside the porous TiO₂ coating layers. It is an intrinsic property of the TiO₂ coatings and determined by the nature of TiO₂ and the coating structures^{1,2}. The internal mass transfer could be evaluated using the Thiele modulus (ϕ) and the internal effectiveness factor (η) as shown in Eqs. 1 and 2, respectively².

$$\phi = \delta \sqrt{\frac{k\tau}{D\varepsilon}} \quad (1)$$

$$\eta = \frac{\tanh \phi}{\phi} \quad (2)$$

where δ is the TiO₂ coating thickness (m), k is the first-order reaction rate constant (s⁻¹), D is the diffusion coefficient of water (2.3×10⁻⁹ m²/s at 20°C³), ε is the TiO₂ coating porosity,

τ is the TiO₂ coating tortuosity calculated using Eq. 3⁴.

$$\tau = \sqrt{\frac{2\varepsilon}{3[1 - 1.209(1 - \varepsilon)^{2/3}]} + \frac{1}{3}} \quad (3)$$

The calculated ϕ and η of the three TiO₂-QOFs are shown in Supplementary Table 5. ϕ of the TiO₂ coating layers decrease from 1.20×10^{-4} in TiO₂-QOF-High to 1.95×10^{-5} in TiO₂-QOF-Low, while all η of the three TiO₂-QOFs equal to 1.00. According to Weisz's criteria, in which the internal mass transfer is neglectable when $\phi < 0.3$ and $\eta \approx 1^5$, the TiO₂ coating layers on the three TiO₂-QOFs have no internal mass transfer limitation.

Therefore, the same external mass transfer and the neglectable internal mass transfer of the TiO₂ coating layers on the three TiO₂-QOFs suggest there is no difference in mass transfer and thus excluding the mass transfer effect on the photocatalytic activity of TiO₂-QOFs.

Supplementary Table 5. Porosity (ε), thickness (δ), tortuosity (τ), Thiele modulus (ϕ) and the internal effectiveness factor (η) of the three TiO₂-QOFs

Name	Porosity (ε)	Thickness (δ : nm)	Tortuosity (τ)	Thiele modulus (ϕ)	Internal effectiveness factor (η)
TiO ₂ -QOF-High	0.622	507.51	1.208	1.20×10^{-4}	1.00
TiO ₂ -QOF-Med	0.710	136.31	1.157	2.89×10^{-5}	1.00
TiO ₂ -QOF-Low	0.896	104.72	1.072	1.95×10^{-5}	1.00

This is added to **the revised Manuscript**:

Lines 276–278: “but not to the mass transfer limitation (Supplementary Section 6.3) or carbamazepine adsorption (Supplementary Fig. 14c) of different TiO₂ coating layers.”

The discussion above is also added to **the revised Supplementary Information Section 6.3**.

Reference:

- [1] Chen, D., Li, F. & Ray, A. K. Effect of mass transfer and catalyst layer thickness on photocatalytic reaction. *AIChE J.* **46**, 1034–1045 (2000).
- [2] Visan, A., Van Ommen, J. R., Kreutzer, M. T. & Lammertink, R. G. H. Photocatalytic reactor design: Guidelines for kinetic investigation. *Ind. Eng. Chem. Res.* **58**, 5349–5357 (2019).
- [3] Barnes, C. J. & Turner, J. V. Isotopic Exchange in Soil Water. *in Isotope Tracers in Catchment Hydrology*. (Elsevier, 1998).
- [4] Ahmadi, M. M., Mohammadi, S. & Hayati, A. N. Analytical derivation of tortuosity and permeability of monosized spheres: A volume averaging approach. *Phys. Rev. E* **83**, 026312 (2011).
- [5] Doran, P. M. Heterogeneous Reactions. *in Bioprocess Engineering Principles*. (Elsevier, 2013).

Comment 2:

It is necessary to observe the leakage of refracted light from the each POF. If no refracted light can be observed, it means that TiO₂ layer is completely absorbed the leakage light and the light absorption process does not affect the difference in photocatalytic activity. Mass transfer and adsorption due to the difference in the thickness of the TiO₂ layer may affect the photocatalytic activity.

Reply:

In response to the reviewer's comment, we performed additional experiments to measure the light irradiance 1 mm away from side surfaces of the three TiO₂-QOFs as shown in Supplementary Table 4. The light irradiance refracted from side surfaces of all the three TiO₂-QOFs at 1 cm and 5 cm along the fiber length is less than 0.10 mW/cm² and 0.01 mW/cm², respectively, which account only for <1.35% and <0.15% of the irradiance launched to TiO₂-QOFs, respectively. Therefore, the light refracted from quartz optical fiber is mostly absorbed by TiO₂ coating layers. Although more light energy was absorbed by TiO₂-QOF-High, 80% of the light energy launched to TiO₂-QOF-High was absorbed at the beginning 2 cm of the fiber, while only about 12% of the light energy was absorbed at the following 4.5 cm. Therefore, the carbamazepine degradation occurred mostly at the beginning 2 cm of the fiber, which limits the photocatalytic activity of TiO₂-QOF-High. On the other hand, the light energy distribution along TiO₂-QOF-Low was more uniform as shown in the original Supplementary Fig. 15. More TiO₂ reactive sites was used for photocatalytic degradation with less light energy. Therefore, TiO₂-QOF-Low showed a higher quantum yield.

This information is added to **the revised Manuscript lines 246–247**: “This dissipated radiant energy was mostly absorbed by the TiO₂ coating layers (details see Supplementary Table 4).”

Supplementary Table 4. Light refracted from side surfaces of TiO₂-QOFs

Name	Light launched to TiO ₂ -QOFs (mW/cm ²)	Light refracted at 1 cm of the TiO ₂ -QOFs (mW/cm ²)	Light refracted at 5 cm of the TiO ₂ -QOFs (mW/cm ²)
TiO ₂ -QOF-High	7.02	0.0942	0.0074
TiO ₂ -QOF-Med	7.02	0.0823	0.0102
TiO ₂ -QOF-Low	7.02	0.0339	0.0099

In response to the comment, we also evaluated the impacts of external mass transfer and internal mass transfer of the TiO₂ coating layers on the photocatalytic performance of different TiO₂-QOFs. The external mass transfer of the three TiO₂-QOFs are the same in the current manuscript because the reactor setup is the same, while the TiO₂ coating layers on the three TiO₂-QOFs have no internal mass transfer limitation (Supplementary Table 5).

These results exclude the difference in mass transfer of the three TiO₂-QOFs affecting the photocatalytic activity.

Tests of the carbamazepine adsorption by TiO₂-QOFs were conducted in the original manuscript Supplementary Fig. 14a, in which carbamazepine concentrations did not change in dark in the presence of TiO₂-QOFs. We also performed additional experiments in response to reviewer's comment to test whether carbamazepine is adsorbed by TiO₂, which confirmed that 5 g/L of TiO₂ suspension cannot adsorb carbamazepine at an initial concentration of 2 μM and other experimental conditions commensurate with our study conditions (Supplementary Fig. 14c).

Supplementary Figure 14c. Carbamazepine (CBZ) adsorbed by TiO₂ suspension. (Conditions: reactor volume = 100 mL, [CBZ] = 2 μM, TiO₂ concentration = 5 g/L)

These are added to **the revised Manuscript:**

Lines 266–268: “Control tests confirmed that carbamazepine was not adsorbed by TiO₂-QOFs in the dark or not degraded when light was launched from LEDs into an uncoated optical fiber (Supplementary Figs. 14a and 14b).”

Lines 276–278: “but not to the mass transfer limitation (Supplementary Section 6.3) or carbamazepine adsorption (Supplementary Fig. 14c) of different TiO₂ coating layers.”

This result is also added to **the revised Supplementary Information Fig. 14c.**

Comment 3:

In photocatalytic decomposition experiments, the change in concentration of CBZ until the adsorption equilibrium is reached should be shown. If it does not adsorb, what is the mechanism by which decomposition is considered to be proceeding?

Reply:

Tests of the carbamazepine adsorption by TiO₂-QOFs were conducted in the original manuscript Supplementary Fig. 14a, in which carbamazepine concentrations did not change in dark in the presence of TiO₂-QOFs. We also performed additional experiments in response to reviewer's comment to test whether carbamazepine is adsorbed by TiO₂, which confirmed that 5 g/L of TiO₂ suspension cannot adsorb carbamazepine at an initial concentration of 2 μM and other experimental conditions commensurate with our study conditions (Supplementary Fig. 14c). Therefore, the decomposition of carbamazepine occurred not on surface but in bulk solution and pores in TiO₂ coating layers. It was attributed to photocatalytic generated hydroxyl radicals (HO•) from TiO₂ coating layers.

These are revised in **the revised Manuscript**:

Lines 45–46: “Upon absorption of light, photocatalysts generate hole-electron (h⁺-e⁻) pairs”,

Lines 82–84: “the interactions between TiO₂ and the evanescent wave energy for the generation of hydroxyl radicals (HO•) to degrade a refractory pollutant (carbamazepine) in bulk solution and pores in TiO₂ coating layers (Fig. 1b)”.

Comment 4:

What is oxidized and reduced in the photocatalytic reaction? Should be shown with a band structure model of the prepared photocatalyst.

Reply:

The redox reactions and band structure model of TiO₂ P25 are added to **the revised Manuscript Fig. 1b**. TiO₂ has a bandgap of 3.2 eV. Upon light irradiation with energy higher or equivalent to the bandgap, TiO₂ is activated and continues to generate hole-electron (h⁺-e⁻) pairs, which are free to migrate. e⁻ migrates to the surface of TiO₂ and interacts with surface adsorbed O₂ to form O₂^{•-}:

O₂^{•-} then transforms to H₂O₂, which then generates HO•:

h⁺ also migrates to the surface and reacts with H₂O and HO⁻, generating HO•:

HO• generated from the photocatalytic processes then degrades carbamazepine:

Figure 1b. Schematic of mechanism of individual TiO₂ nanoparticles (within orange dashed lined box) and as a collection of photocatalysis on the surface of the optical fiber that interact with the evanescent wave energy to degrade carbamazepine by photocatalytic generated hydroxyl radicals (HO•) in water.

Reference:

[1] Pichat, P. *Photocatalysis and Water Purification: From Fundamentals to Recent Applications* (Wiley-VCH press, Weinheim, 2013).

Comment 5:

The photocatalytic decomposition data presented by the authors showed the highest activity in POF with the least photocatalytic coating. Is the thinner the coating of the photocatalyst layer, the higher the activity? Is there a peak in the relationship between photocatalytic amount and activity?

Reply:

a. Regarding the question of “Is the thinner the coating of the photocatalyst layer, the higher the activity”.

According to our energy balance model, the TiO₂ layer structure parameters (p and z_a) are both important to control the ratio of refracted light to evanescent waves, while the TiO₂ coating thickness is not critical.

This is added to **lines 258–260 of the revised Manuscript**: “The above modeling results suggest the TiO₂ layer structure parameters (p and z_a) are important to control the light energy within TiO₂-QOFs, while the TiO₂ coating thickness is not critical.”

b. Regarding the question of “Is there a peak in the relationship between photocatalytic amount and activity”.

We fabricate two more TiO₂-QOFs, including TiO₂-QOF-Low’’ at p of 0.018 and z_a of 139.50 nm and TiO₂-QOF-Low’ at p of 0.026 and z_a of 127.97 nm, using different coating conditions as shown in Supplementary Table 6. These two TiO₂-QOFs are then compared with TiO₂-QOF-High, TiO₂-QOF-Med, and TiO₂-QOF-Low. As shown in Supplementary Table 6, the carbamazepine degradation rate constants increased with increasing p and decreasing z_a from TiO₂-QOF-Low’’ to TiO₂-QOF-Low, but remain the same with further increasing p and decreasing z_a from TiO₂-QOF-Low to TiO₂-QOF-High. A turning point exists where the degradation rate constants reach a plateau. On the other hand, quantum yields of carbamazepine degradation by the UV irradiated TiO₂-QOF-Low was the highest, at 0.0248. Increasing p and decreasing z_a or decreasing p and increasing z_a resulted in a decrease in quantum yields.

Supplementary Table 6. Dip-coating conditions and TiO₂ coating parameters of the five TiO₂-QOFs

Name	Dip-coating conc. (mg/L)	Dipping duration (min)	Coating cycles	p	z_a (nm)	Degradation rate constants (h ⁻¹)	Quantum yields
TiO ₂ -QOF-High	10,000	0.5	5	0.528	7.725	0.237	0.0189
TiO ₂ -QOF-Med	10,000	0.5	1	0.206	52.874	0.229	0.0195
TiO ₂ -QOF-Low	40	60	1	0.034	114.295	0.240	0.0248
TiO ₂ -QOF-Low’	40	30	1	0.026	127.973	0.120	0.0138
TiO ₂ -QOF-Low’’	20	30	1	0.018	139.500	0.076	0.0093

This information is added to **the revised Manuscript**:

Lines 278–281: “We then fabricated two new TiO₂-QOFs, i.e., TiO₂-QOF-Low’’ at p of 0.018 and z_a of 139.50 nm and TiO₂-QOF-Low’ at p of 0.026 and z_a of 127.97 nm. Both degradation rate constants and quantum yields decreased with decreasing p and increasing z_a (Supplementary Table 6).”

The discussion above is also added to **the revised Supplementary Information**:

Section 6.4: “We fabricate two more TiO₂-QOFs, including TiO₂-QOF-Low’’ at p of 0.018 and z_a of 139.50 nm and TiO₂-QOF-Low’ at p of 0.026 and z_a of 127.97 nm, using different coating conditions as shown in Supplementary Table 6. These two TiO₂-QOFs are then compared with TiO₂-QOF-High, TiO₂-QOF-Med, and TiO₂-QOF-Low. As shown in Supplementary Table 6, the carbamazepine degradation rate constants increased with increasing p and decreasing z_a from TiO₂-QOF-Low’’ to TiO₂-QOF-Low, but remained

the same with further increasing p and decreasing z_a from TiO₂-QOF-Low to TiO₂-QOF-High. A turning point exists where the degradation rate constants reach a plateau. On the other hand, quantum yields of carbamazepine degradation by the UV irradiated TiO₂-QOF-Low was the highest, at 0.0248. Increasing p and decreasing z_a or decreasing p and increasing z_a resulted in a decrease in quantum yields. ”

Comment 6:

What solvent was used for dip coating of TiO₂? During the photocatalytic reaction, is the coating come off without calcination?

Reply:

TiO₂ suspension was prepared by dispersing TiO₂ in double deionized water (18.2 MΩ-cm), and no organic solvents were utilized. This information is added to **lines 324–325 of the revised Manuscript**: “TiO₂ suspension was prepared by dispersing TiO₂ (P25) in double deionized water (18.2 MΩ-cm), and no organic solvents were utilized.”

In response to the reviewer’s comment, we performed an indirect test to demonstrate that TiO₂ coating is stable during the reaction. Tests are conducted to evaluate the radiant energy dissipation and carbamazepine degradation rate constants of TiO₂-QOF-Low for 3 cycles. For each cycle, TiO₂-QOF-Low was immersed in the carbamazepine containing solution under stirring for 4 h. After each cycle, the used TiO₂-QOF-Low was dried before running the next cycle. Supplementary Fig. 17 shows that both radiant energy dissipation and carbamazepine degradation rate constants remain unchanged for the 3 cycles of testing, suggesting TiO₂ coating is stable during the reaction.

Supplementary Figure 17. Radiant energy dissipation and carbamazepine degradation rate constants of TiO₂-QOF-Low for 3 cycles. (Conditions: light intensity = 7.02 mW/cm², wavelength = 365 nm, TiO₂ coating length = 6.5 cm, irradiation duration = 4 h)

These are added to **the revised Manuscript:**

Lines 301–303: “The TiO₂ coating in TiO₂-QOF-Low is also stable, because the radiant energy dissipation and degradation rate constants remained unchanged for 3 cycles of carbamazepine degradation (Supplementary Fig. 17).”

This information is also added to **the revised Supplementary Information:**

Section 6.7: “We performed an indirect test to demonstrate that TiO₂ coating is stable during the reaction. Tests are conducted to evaluate the radiant energy dissipation and carbamazepine degradation rate constants of TiO₂-QOF-Low for 3 cycles. For each cycle, TiO₂-QOF-Low was immersed in the carbamazepine containing solution under stirring for 4 h. After each cycle, the used TiO₂-QOF-Low was dried before running the next cycle. Supplementary Fig. 17 shows that both radiant energy dissipation and carbamazepine degradation rate constants remain unchanged for the 3 cycles of testing, suggesting TiO₂ coating is stable during the reaction.”

Comment 7:

TiO₂-QOF-High, which has a large amount of deposition, should have more TiO₂ far from the optical fiber surface than other POFs. Why is z_a of TiO₂-QOF-Low the largest?

Reply:

We appreciate the comment. We revised the definition of z_n on **lines 165–168 of the revised manuscript** as “ z_n is the normalized distance between the fiber surface and TiO₂ nanoparticles at a TIR spot, which is equal to the average distance from the center of an evanescent field to the closest surrounding TiO₂ nanoparticles within the evanescent field”. While z_a equals to the average value of z_n at all TIR spots along the fiber length. The definition of z_a indicates that evanescent wave energy dissipates when they reach the closest TiO₂ nanoparticles, while they return to the fiber if not reach TiO₂. Our model shows that z_a is largest in TiO₂-QOF-Low and smallest in TiO₂-QOF-High among the three TiO₂-QOFs. Although TiO₂-QOF-High has a large amount of TiO₂ deposition, these TiO₂ nanoparticles are far from the optical fiber surface and they do not absorb evanescent waves. The above information is revised in **the revised manuscript lines 251–254**: “The value of z_a in TiO₂-QOF-High (7.7 nm) is much smaller than the value of λ from quartz to water (50–120 nm) (Supplementary Fig. 11), suggesting evanescent wave energy dissipates when they reach the closest TiO₂ nanoparticles and a negligible amount of them returned to the optical fibers to give a highest ratio of $E_{E,dis}$ ’ to $E_{E,g}$.”

REVIEWER COMMENTS

Reviewer #1 (Remarks to the Author):

Even though the authors have clarified some issues in their rebuttal and improved the manuscript in many aspects, I am still very sceptical about the proposed superiority of their system.

The way I understand it, the authors merely shift the light losses from refraction to transmission. Since the former is not accurately determined (see below) and deducted from the absorbed photon flux but the latter is, this results in their supposedly higher quantum yield. The authors claim that the transmission losses can be counteracted by just using a longer fibre length but also the refraction losses are strongly attenuated by using fibre bundles. So in either case, there is no real advantage.

The only potential advantage would be that the light is indeed more evenly dissipated and this may prevent local photon oversaturation and its associated efficiency losses. However, the authors do not provide any evidence for oversaturation in the case of the highly coated fibers so this is pure speculation.

Overall, the paper is quite intriguing and well written. However, as there seems to be no real benefit of the new system the impact is not sufficiently high to warrant publication in Nature Communications.

On the determination of the refracted light:

The methodology of determining the refraction losses is highly questionable. If the irradiance is measured 1 mm away from the fiber, it is already quite dispersed. In fact, the total area illuminated at 1mm distance to the fiber is 780 times larger than the cross section of the fiber at 6.5 cm length and even 3120 times larger for 26 cm length. This needs to be taken into account when comparing irradiances. As such I suspect the actual reflection losses are many times the value stated by the authors and may therefore not be so easily discounted. An accurate determination would entail an integration of the total power (or photon flux) emitted by the fibre over its entire length (either experimentally or computationally based on few measured data points).

Their data also suggests that the highly coated fibre refracts significantly more light (almost three times) than the low coated one. The higher refraction losses of the former may readily explain the differences in the observed (supposed) quantum yield.

Reviewer #2 (Remarks to the Author):

This manuscript has been well revised. Thus I recommend its acceptance without any change

Title: Evanescent waves modulate energy efficiency of photocatalysis within nano-TiO₂ coated quartz optical fibers illuminated using LEDs

Authors: Yinghao Song, Li Ling, Paul Westerhoff, Chii Shang

Manuscript ID: NCOMMS-20-44004B

Before responding to the comments of the reviewers, the authors would like to express our deepest appreciation for their time and effort in reviewing this manuscript.

Responses to comments of Reviewer 1:

Comment 1:

Even though the authors have clarified some issues in their rebuttal and improved the manuscript in many aspects, I am still very skeptical about the proposed superiority of their system.

Overall, the paper is quite intriguing and well written. However, as there seems to be no real benefit of the new system the impact is not sufficiently high to warrant publication in Nature Communications.

Reply:

Thank you for recognizing the intriguing parts of our paper. Below is a discussion that lays out the significance of the work, and addresses the specific point about superiority of our system.

In last 5 years, over 60,000 journal papers were published about photocatalytic processes as recorded in Web of Science database. These publications covered a wide range of fields including energy production, pollutant remediation, organic synthesis, etc. Most of these papers focused on developing new catalytic materials to enhance the quantum yields of photocatalytic processes. An important but often overlooked aspect to limiting use photocatalytic processes into engineering practice is the influence of photocatalytic reactor design, rather than material properties alone. We highlighted this in our recent review article¹, published in late 2018 which has over 15,000 downloads and already over a hundred citations. To address the barrier of reactor design rather than material properties, we show how quantum yields are effectively increased more significantly by managing the way light reaches the photocatalyst, than recent improvements in catalyst materials themselves.

Here, we developed a novel coating strategy to create a new type of fixed-film reactor that left 3.4% of the optical fiber surface in direct contact with TiO₂ and, on-average, 114.3 nm interspace distance between the fiber surface and the coated TiO₂ layers. This strategy successfully reduced refraction, generated evanescent waves in the TiO₂-QOF, and thus allowed even dissipation of light along TiO₂-QOF to prevent oversaturation of the light delivered to the fiber and its associated efficiency losses (details see reply to Comment 2). As a result, this strategy saves 23% of the radiant energy delivered to the TiO₂-QOF without compromising carbamazepine degradation. The saved energy could be utilized by extending the TiO₂ coating length to further enhance pollutant degradation and quantum yields. The novel TiO₂-QOF enables the design of a photocatalytic reactor that uses 77% less mass of photocatalysts but achieves up to 96% improvement in quantum yields compared with reactors built of fibers with densely coated TiO₂ layers, mainly because optimizing evanescent waves to fully use transmitted light in longer TiO₂-QOF-Low is more efficient than harvesting refracted light by using TiO₂-QOF-High bundles (details see reply to Comments 3). Moreover, such a strategy of managing evanescent wave energy is

applicable to all photocatalysts and thus emerges as an important cost consideration especially when contemplating expensive photocatalysts. By replacing TiO₂ to photocatalysts with different functions, for example, hydrogen production or organic synthesis, an improvement of up to 96% in quantum yields will significantly reduce the cost of the processes. In addition to considering only refracted light in our optical fiber reactor, another key insight is the separation of photocatalyst activation by refracted light versus evanescent wave energy to activate the photocatalyst. We report for the first time here the relative importance of both mechanisms, as well as how to modulate their relative importance.

A new platform can also be established using novel photocatalyst-coated optical fibers to allow evanescent wave energy to activate photocatalysts. This new platform can quantify performance of photocatalysts more precisely by minimizing the radiation scattering by photocatalysts and water parameters, preventing aggregation of photocatalysts to maximize the interaction between reactive sites and target compounds, and allowing easier and more accurate quantification of photons absorbed by photocatalysts.

We believe these benefits grant the significance and high impacts of this manuscript to warrant its publication in *Nature Communications*.

Reference:

1. Loeb, S. K. et al. The technology horizon for photocatalytic water treatment: Sunrise or sunset? *Environ. Sci. Technol.* **53**, 2937–2947 (2019).

These are revised from **the latest revised Manuscript**:

Former text in Lines 305–317 “We developed a novel coating strategy that left 3.4% of the optical fiber surface in direct contact with TiO₂ and, on-average, 114.3 nm interspace distance between the fiber surface and the coated TiO₂ layers. This strategy successfully reduced refraction, generated evanescent waves in the TiO₂-QOF, and saved 23% of the radiant energy delivered to the TiO₂-QOF without compromising carbamazepine degradation. The saved energy could be utilized by extending the TiO₂ coating length to further enhance pollutant degradation and quantum yields. This coating strategy also enabled the use of fewer photocatalysts, which emerges as an important cost consideration when contemplating other novel visible light active materials. Because similar optical fiber coating modulation is likely to increase quantum yields using higher wattage and lower-cost visible light LEDs or even the mass abundant sunlight with visible light photocatalysts, our findings on the ability to manage evanescent wave energy by controlling patchiness and interspace distance between optical fiber surfaces and photocatalysts will have broad impacts on the next generation photocatalytic reactor design.” was revised as **Lines 312–341 in the newly revised Manuscript**:

“Photocatalytic processes have broad application in water and air treatment, energy production, organic synthesis, and other fields. Less than 1 in 100 papers address the critical barrier to making photocatalytic reactors more effective, namely light and energy management; the other papers focus largely on discovery of new or incremental

improvement in existing photocatalyst materials¹⁰. An important but often overlooked aspect to limiting use photocatalytic processes into engineering practice is the influence of photocatalytic reactor design, rather than material properties alone. To address the barrier of reactor design rather than material properties we show how quantum yields are effectively increased more significantly by managing the way light reaches the catalyst, than recent improvements in catalyst materials themselves. In addition to considering only refracted light in our optical fiber reactor, another key insight was the separation of photocatalyst activation by refracted light versus evanescent wave energy to activate the photocatalyst. We report for the first time here the relative importance of both mechanisms, as well as how to modulate their relative importance. Here, we developed a novel coating strategy that left 3.4% of the optical fiber surface in direct contact with TiO₂ and, on-average, 114.3 nm interspace distance between the fiber surface and the coated TiO₂ layers. This strategy successfully reduced refraction, generated evanescent waves in the TiO₂-QOF, and thus allowed even dissipation of light along TiO₂-QOF to prevent oversaturation of the light delivered to the fiber and its associated efficiency losses. The novel TiO₂-QOF enables the design of a photocatalytic reactor that uses 77% less mass of photocatalysts but achieves up to 96% improvement in quantum yields compared with reactors built of fibers with densely coated TiO₂ layers, mainly because optimizing evanescent waves to fully use transmitted light in longer TiO₂-QOF-Low is more efficient than harvesting refracted light by using TiO₂-QOF-High bundles. Moreover, such a strategy of managing evanescent wave energy is applicable to all photocatalysts and thus emerges as an important cost consideration especially when contemplating expensive photocatalysts.

A new platform can also be established using novel photocatalyst-coated optical fibers to allow evanescent wave energy to activate photocatalysts. This new platform can quantify performance of photocatalysts more precisely by minimizing the radiation scattering by photocatalysts and water parameters, preventing aggregation of photocatalysts to maximize the interaction between reactive sites and target compounds, and allowing easier and more accurate quantification of photons absorbed by photocatalysts. ”

Comment 2:

The only potential advantage would be that the light is indeed more evenly dissipated and this may prevent local photon oversaturation and its associated efficiency losses. However, the authors do not provide any evidence for oversaturation in the case of the highly coated fibers so this is pure speculation.

Reply:

We appreciate that the reviewer raised that we need to better articulate advantages of TiO₂-QOF-low regarding to how we prepared fibers to enable light to be more evenly dissipated along the fiber and prevents local photon oversaturation and its associated efficiency losses. In response to the reviewer's comment, additional experiments and data analysis were conducted. These are show in SI, as follows:

In Supplementary Fig. 15a, we show the simulated radiant flux dissipated at each coating section at a length of 1 cm along TiO₂-QOFs as calculated from the energy balance model. Approximately 80% of the light launched to TiO₂-QOF-High was dissipated at the first 2-cm section, while only 12% of which was dissipated at the following sections. In contrast, only 43% of the light launched to TiO₂-QOF-Low was dissipated at the first 2-cm section and light was more evenly dissipated along the fiber.

The large amount of light dissipated at the beginning section of TiO₂-QOF-High resulted in local photon oversaturation and caused efficiency losses. This was proved by an additional experiment, which shows the degradation rate constants of carbamazepine and quantum yields by the UV-irradiated TiO₂-QOF-High at a coating length of 1 cm as a function of radiant flux dissipation (Supplementary Fig. 15b). With increasing radiant flux dissipation from 3 to 36 μW , the degradation rate constants of carbamazepine by 1-cm TiO₂-QOF-High increased by 1.8 times, while the quantum yields showed a significant drop by 77%. Therefore, the radiant flux dissipated at the first 2-cm section of TiO₂-QOF-High at 44 μW contributed to over 75% of carbamazepine degradation but with a low quantum yields of 0.014, while only 25% of carbamazepine degradation was attributed to the last 4-cm of TiO₂-QOF-High. The local photon oversaturation and its high efficiency losses at the first 2-cm section cause the low overall quantum yield of 6-cm TiO₂-QOF-High. In contrast, TiO₂-QOF-Low has a more evenly dissipated radiant flux of 2 to 15 μW , which grant TiO₂-QOF-Low higher quantum yields in degrading carbamazepine.

The novel fiber coating strategy we discovered and reported in this manuscript modulates the coating layer structures to generate higher portions of evanescent waves. The generation of evanescent waves results in even dissipation of light in TiO₂-QOFs and thus prevents local light oversaturation and its associated efficiency losses.

Supplementary Figure 15. (a) Radiant flux dissipated at each coating section at a length of 1 cm along TiO₂-QOF-High and TiO₂-QOF-Low at an incident light intensity of 7.0 mW/cm²; (b) carbamazepine degradation by TiO₂-QOF-High at a coating length of 1mm as a function of radiant flux dissipation. (Conditions: light wavelength = 365 nm, initial carbamazepine concentration = 2 μM, irradiation duration = 4 h, reactor volume = 23 mL)

This is added to **the newly revised Manuscript Lines 275–278**:

“The evanescent waves generated in TiO₂-QOF-Low allows light to be evenly dissipated along the fiber, which prevents local photon oversaturation at the beginning sections of TiO₂-QOFs and thus reduces its associated efficiency losses (Supplementary Section 6.2).

These are revised from **the latest revised Manuscript**:

Former text in Lines 275–277: “but not to the mass transfer limitation (Supplementary Section 6.3) or carbamazepine adsorption (Supplementary Fig. 14c) of different TiO₂ coating layers.” was revised as **Lines 278–280 in the newly revised Manuscript:**

“We also proved that the improved quantum yield was not attributed to the mass transfer limitation (Supplementary Section 6.3) or carbamazepine adsorption (Supplementary Fig. 14c) of different TiO₂ coating layers.”

We delete the text in **the latest revised Manuscript Lines 270–271:** “The hypothesis about why k was not compromised is shown in Supplementary Section 6.2.”

The information about the even dissipation of light in TiO₂-QOF-Low prevents local photon oversaturation is revise in **the revised Supplementary Information Section 6.2.**

Comment 3:

The way I understand it, the authors merely shift the light losses from refraction to transmission. Since the former is not accurately determined (see below) and deducted from the absorbed photon flux but the latter is, this results in their supposedly higher quantum yield. The authors claim that the transmission losses can be counteracted by just using a longer fiber length but also the refraction losses are strongly attenuated by using fiber bundles. So in either case, there is no real advantage.

Reply:

To demonstrate the advantage of our system, we conducted additional experiments to further confirm that the use of the transmitted light by using longer TiO₂-QOF-Low is more efficient than the use of refracted light by using TiO₂-QOF-High bundles. We conducted additional experiments in which seven TiO₂-QOFs were bundled together and used for carbamazepine degradation, as shown in Supplementary Figs. 17 and 18. The conclusion is that optimizing evanescent waves to fully use transmitted light in longer TiO₂-QOF-Low is 44–96% more efficient than harvesting refracted light by using TiO₂-QOF-High bundles depending on fiber spacings (1 mm to 7 mm). Meanwhile, TiO₂-QOF-Low uses 77% fewer photocatalysts than TiO₂-QOF-High. Below are details on these additional experimental results which are integrated into the manuscript:

As shown in Supplementary Fig. 17a, a hexagonal arrangement for a TiO₂-QOF bundle which consists of one TiO₂-QOF in the center and six TiO₂-QOFs at the edge was proposed. The hexagonal arrangement is the closest packing that allows TiO₂-QOFs to utilize the most refracted light out of fibers. The minimum distance between two TiO₂-QOF surfaces, either from the TiO₂-QOF in the center to the ones at the edge or between those at the edge, are defined as the fiber spacing (S) in the bundle. The S was set as 1, 3, 5 and 7 mm in the supplementary tests. The TiO₂-QOF bundle was installed in a tubular reactor of an inner length of 65 mm and an inner diameter of 24 mm. All seven TiO₂-QOFs have a coating length of 6.5 cm. Light was only allowed to be launched from a UV-LED to the TiO₂-QOF in the center of the bundle, i.e., the one centered in the axial of the reactor.

Supplementary Fig. 17b shows the comparisons in carbamazepine degradation rates (r) by a single fiber of TiO₂-QOF-High, a single fiber of TiO₂-QOF-Low, a bundle of TiO₂-QOF-High, and a bundle of TiO₂-QOF-Low irradiated by one UV-LED at a UV intensity of 7.02 mW/cm². The S in the two bundles were 1 mm. The carbamazepine degradation rate by a 6.5-cm TiO₂-QOF-High (r_{c-High}) was 0.0065 $\mu\text{mole h}^{-1}$. Under the same experimental condition, the carbamazepine degradation rate by a 6.5 cm TiO₂-QOF-Low was also 0.0065 $\mu\text{mole h}^{-1}$. But to fully utilize the incident light launched into the single TiO₂-QOF-Low, it shall be extended to 26 cm. The 26 cm TiO₂-QOF-Low thus showed a carbamazepine degradation rate (r_{c-Low}) of 0.0127 $\mu\text{mole h}^{-1}$, which was 97% higher than r_{c-High} (Manuscript Fig. 5d). By bundling seven 6.5-cm TiO₂-QOF-High together using our proposed hexagonal arrangement, the carbamazepine degradation rate by a bundle of TiO₂-QOF-High (r_{High}) was 0.011 $\mu\text{mole h}^{-1}$, which was 67% higher than r_{c-High} . Moreover, by bundling seven 26-cm TiO₂-QOF-Low, the carbamazepine degradation rate was further improved to 0.015 $\mu\text{mole h}^{-1}$, as calculated by adding up the carbamazepine degradation rates by a 6.5-cm TiO₂-QOF-Low bundle and that by the extended 19.5 cm portion of the

26-cm TiO₂-QOF-Low in the center. The degradation rate by a 26-cm TiO₂-QOF-Low bundle (r_{Low}) of 0.015 $\mu\text{mole h}^{-1}$ is thus about 1.4 times as high as r_{High} . In TiO₂-QOF bundles, r was attributed to one TiO₂-QOF in the center irradiated directly by a UV-LED and six TiO₂-QOFs at the edge irradiated by refracted light from the one in the center. Therefore, r_{High} and r_{Low} can be expressed as Eqs. 1 and 2, respectively.

$$r_{High} = r_{c-High} + 6r_{e-High} \quad (1)$$

$$r_{Low} = r_{c-Low} + 6r_{e-Low} \quad (2)$$

where r_{e-High} and r_{e-Low} are the carbamazepine degradation rate attributed to each of the six TiO₂-QOF-High at the edge and each of the six TiO₂-QOF-Low at the edge, respectively. Besides, since all the four TiO₂-QOF systems received the same UV intensity, their apparent quantum yields followed the same trend as their degradation rates.

As the S affects the amount of refracted light received by the TiO₂-QOFs at the edge and thus the carbamazepine degradation rate by a TiO₂-QOF bundle. r as a function of S were examined and shown in Supplementary Fig. 17c using the above-mentioned experimental setup. With an increase in S from 1 to 7 mm, both r_{High} and r_{Low} decreased proportionally. By subtracting r_{c-High} from r_{High} and r_{c-Low} from r_{Low} , r_{e-High} and r_{e-Low} as a function of S were obtained as shown in Eqs. 3 and 4, respectively. Eqs. 3 and 4 suggest both r_{e-High} and r_{e-Low} decrease with increasing S .

$$r_{e-High} = \frac{-0.0007S + 0.005}{6} \quad (3)$$

$$r_{e-Low} = \frac{-0.0003S + 0.0023}{6} \quad (4)$$

Supplementary Figure 17. (a) Schematics of a hexagonal arrangement for a TiO₂-QOF bundle which consists of one TiO₂-QOF in the center and six TiO₂-QOFs at the edge for carbamazepine degradation; (b) the comparison in carbamazepine degradation rates by a single fiber of TiO₂-QOF-High (6.5 cm), a single fiber of TiO₂-QOF-Low (26 cm), a bundle of TiO₂-QOF-High (6.5 cm), and a bundle of TiO₂-QOF-Low (26 cm) irradiated by a UV-LED; (c) the carbamazepine degradation rates by a TiO₂-QOF-Low bundle irradiated by one UV-LED (r_{Low}) and that by a TiO₂-QOF-High bundle irradiated by one UV-LED (r_{High}) as a function of fiber spacings (S). (Conditions: light intensity = 7.0 mW/cm², wavelength = 365 nm, initial carbamazepine concentration = 2 μ M, irradiation duration = 4 h).

However, in practice, light will be delivered to multiple TiO₂-QOFs in the bundle, rather than a single TiO₂-QOF. Therefore, the carbamazepine degradation rates by (i) a bundle of TiO₂-QOF-Low consisting of seven 26-cm TiO₂-QOF-Low each irradiated by one UV-LED (r'_{Low}), and (ii) a bundle of TiO₂-QOF-High consisting of seven 6.5-cm TiO₂-QOF-High each irradiated by one UV-LED (r'_{High}), were simulated using the data we obtained from Supplementary Fig 17c. The TiO₂-QOF in the center receives light launched from a UV-LED and refracted light from the surrounding six TiO₂-QOFs at $S = x$ (Supplementary Fig. 18a). Each of the six TiO₂-QOFs at the edge receives light launched from a UV-LED, refracted light from the surrounding three TiO₂-QOFs at $S = x$, and refracted light from two TiO₂-QOFs at a further distance $S' = [(x+1)3^{0.5}-1]$ (Supplementary Fig. 18b). Therefore, the overall carbamazepine degradation rate of seven TiO₂-QOFs each irradiated by one UV-LED (r') were calculated by summing the carbamazepine degradation rate by each of the seven TiO₂-QOFs as shown in Eq. 5.

$$r' = \left(r_c + 6r_{e|S=x} \right) + 6 \left(r_c + 3r_{e|S=x} + 2r_{e|S'=\sqrt{3}(x+1)-1} \right) \quad (5)$$

The r'_{Low} and r'_{High} as a function of S were then calculated and shown in Supplementary Fig. 18c. Both r'_{Low} and r'_{High} were the highest at an S of 1 mm. At such a small S , r'_{Low} is 44% higher than r'_{High} . This result shows TiO₂-QOF-Low which generates high quantity of evanescent waves to activate TiO₂ is more efficient to degrade carbamazepine compared with TiO₂-QOF-High even when they are bundled together. Nonetheless, such compact arrangement of TiO₂-QOFs with an S of 1 mm cannot guarantee uniform mixing as the reactor scales up when installing more TiO₂-QOFs and may compromise the degradation rates and apparent quantum yields. A larger S is thus required. With increasing S from 1 to 7 mm, the differences between r'_{Low} and r'_{High} increase from 44% to 96% (Supplementary Fig. 18c). The advantage of TiO₂-QOF-Low becomes more significant compared with TiO₂-QOF-High at larger S . Besides, a 26-cm TiO₂-QOF-Low bundle uses 77% less mass of photocatalysts compared with that used in a 6.5-cm TiO₂-QOF-High bundle.

These results show that optimizing evanescent waves to fully use transmitted light in longer TiO₂-QOF-Low is 44–96% more efficient than harvesting refracted light by using TiO₂-QOF-High bundles depending on fiber spacings (1 mm to 7 mm). By controlling surface patchiness and distance between fiber surface and photocatalyst coating layers, TiO₂-QOF-Low not only prevents light oversaturation and its associated efficient losses (as demonstrated in Comment 2), but also reduces light wasted by refraction and increase surface reactive sites. These features make TiO₂-QOF-Low more energy-efficient to degrade pollutants.

Supplementary Figure 18. Schematics of the cross-section of a TiO₂-QOF bundle to simulate (a) the carbamazepine degradation rate by the TiO₂-QOF in the center, which equals to the sum of the carbamazepine degradation rate by this TiO₂-QOF irradiated by a UV-LED (r_c) and that by this TiO₂-QOF irradiated by refracted light from the surrounding six TiO₂-QOFs at a fiber spacing (S) of x ($6 \times r_{e|S=x}$), and (b) the carbamazepine degradation rate by the TiO₂-QOF at the edge, which equals to the sum of r_c , $3 \times r_{e|S=x}$, and $2 \times r_{e|S=(x+1)3^{0.5}-1}$; (c) the carbamazepine degradation rates by the TiO₂-QOF-Low bundle each irradiated by one UV-LED (r'_{Low}) and that by the TiO₂-QOF-High bundle each irradiated by one UV-LED (r'_{High}) as a function of S . (Conditions: light intensity of a UV-LED = 7.0 mW/cm², wavelength = 365 nm, initial carbamazepine concentration = 2 μM, irradiation duration = 4 h).

This is added to **the newly revised Manuscript Lines 298–305**:

“Even if refracted light emitted from TiO₂-QOF-High can be utilized by bundling TiO₂-QOFs together, optimizing evanescent waves to fully use transmitted light in longer TiO₂-QOF-Low is 44–96% more efficient (in carbamazepine degradation rates and apparent quantum yields) than harvesting refracted light by using TiO₂-QOF-High bundles (detailed see Supplementary Section 6.7). By controlling surface patchiness and distance between fiber surface and photocatalyst coating layers, TiO₂-QOF-Low not only prevent light oversaturation and its associated efficient losses, but also reduces light wasted by refraction and increase surface reactive sites. These features make TiO₂-QOF-Low more energy-efficient to degrade pollutants.”

These are revised from **the latest revised Manuscript**:

Former text in Lines 282–289: “When assessing the apparent quantum yield (moles of carbamazepine degraded per mole of photons launched to optical fibers), both the light energy absorbed by TiO₂ and the transmission losses should also be considered. The apparent quantum yield of TiO₂-QOF-Low at 6.5 cm shows no superiority compared with TiO₂-QOF-High or TiO₂-QOF-Med. However, its apparent quantum yield can be further increased by bundling multiple TiO₂-QOFs with a single LED as reported before³⁷ or, as we show here, extending the length of TiO₂-QOF-Low to utilize the returned radiant energy and increase the photocatalytic reactive sites.” was revised as **Lines 285–288 in the newly revised Manuscript**:

“The quantum yield of TiO₂-QOF-Low can be further increased by extending its length to fully utilize the returned radiant energy of evanescent waves in TiO₂-QOF-Low and to increase its photocatalytic reactive sites.”

These additional modifications were made in **the revised Manuscript**:

Former text in Lines 294–297: “therefore, the 26-cm TiO₂-QOF-Low was estimated to achieve 2× degradation rate constants and 2× apparent quantum yields compared with TiO₂-QOF-High at a coating length of 6.5 cm (Supplementary Section 6.6).” was revised as **Lines 293–296 in the newly revised Manuscript**:

“therefore, the 26-cm TiO₂-QOF-Low was estimated to achieve 2× degradation rate constants and 2× apparent quantum yields (moles of carbamazepine degraded per moles of photons launched to optical fibers) compared with TiO₂-QOF-High at a coating length of 6.5 cm (Supplementary Section 6.6).”

The information contained in response to Comment 3 is added to **the revised Supplementary Information Section 6.7**.

Comment 4:

On the determination of the refracted light:

The methodology of determining the refraction losses is highly questionable. If the irradiance is measured 1 mm away from the fiber, it is already quite dispersed. In fact, the total area illuminated at 1 mm distance to the fiber is 780 times larger than the cross section of the fiber at 6.5 cm length and even 3120 times larger for 26 cm length. This needs to be taken into account when comparing irradiances. As such I suspect the actual reflection losses are many times the value stated by the authors and may therefore not be so easily discounted. An accurate determination would entail an integration of the total power (or photon flux) emitted by the fiber over its entire length (either experimentally or computationally based on few measured data points).

Their data also suggests that the highly coated fiber refracts significantly more light (almost three times) than the low coated one. The higher refraction losses of the former may readily explain the differences in the observed (supposed) quantum yield.

Reply:

Authors have shown that the harvest of refraction losses of light by using bundles of highly coated fibers (TiO₂-QOF-High) is less efficient than recovering transmission losses of light by using longer fibers with controlled TiO₂ coating morphology (TiO₂-QOF-Low) in replies to Comment 2 and 3. Therefore, it is not necessary to measure the refracted light emitting out from the TiO₂-QOFs.

We delete the text in **the latest revised Manuscript:**

Lines 246–247: “This dissipated radiant energy was mostly absorbed by the TiO₂ coating layers (details see Supplementary Table 4).”

We delete the text in **the latest revised Supplementary Information Section 5.7:** Light emitted from side surfaces of TiO₂-QOFs.

REVIEWERS' COMMENTS

Reviewer #1 (Remarks to the Author):

The authors have now supplied additional information which demonstrates that indeed their system can be superior by minimizing local light oversaturation. With this, all points raised by me have now been addressed.

In light of the potential impact of this novel reaction system, I recommend the article be published.